# Western Message Petroglyphs: A Faux Indian Picture-Writing Project in the American West

Leigh Marymor 

Museum of Northern Arizona, Flagstaff, AZ 86001, USA; mleighm33@gmail.com

**Abstract:** The term "Western Message Petroglyphs" (WMPs) refers to a number of petroglyph sites found scattered among eight western states that are recognized by their shared image content and layout. The imagery is drawn largely from a mash-up of late historic Native American sign-gesture language and picture-writing traditions inter-mixed with pan-cultural imagery from around the world. An increasing number of sites that fit this mold have been reported over the past 85 years or so, currently numbering 39 in all. There is no question that these sites date to post-European contact based on images in some panels that depict Euro-American cultural content (e.g., western-style house, rifle, whiskey keg, horse, etc.). The post-contact era is also apparent in the method used in rendering the engraved images evidenced by the smooth angular lines and chisel strikes produced by metal tools. This paper focuses on narrowing the time frame for these sites based on two additional streams of evidence. First, patterned associations with historic landscape settings tied to the era of western expansion bind the sites together into a coherent whole and set a floor for their oldest probable dates. An example of four sites located in Utah and Arizona illustrates their connection to the last quarter of the nineteenth century and the opening years of the twentieth. Secondly, a study of their imagery supports the proposed dates by revealing a "smoking gun" for the source of many of the individual icons. An example of the methodology used to translate a Western Message Petroglyph panel is described, and a profile of the central author who appears to have acted with a small group of others is suggested in order to aid in the search for this person(s) in the historic record of the American West.

**Keywords:** Western Message Petroglyphs; Native American picture-writing; pan-cultural symbols; "Faux-Indian" historic inscriptions



## 1. Introduction

On 18 April 2020 Stewart Lasseter Wyld posted a photograph that he titled "Ash Fork, AZ; Nice nachwach!" to *Petroglyphs, Pictographs, and Rock Art of North America*, a public *Facebook* group primarily interested in Indigenous cultural markings on rock surfaces located in natural landscape settings. His interest had been drawn to an engraved element comprised of two inter-locking "C" shapes that he recognized as a depiction of the Hopi "Nachwach" (Nakwach) symbol, a traditional symbol representing friendship, or perhaps brotherhood. What Wyld was not aware of at the time he posted the image was that he had stumbled upon a previously unreported example of an esoteric historic petroglyph tradition known as Western Message Petroglyphs (aka WMPs; Modern Petrography; Mystery Glyphs). With his Internet posting, the corpus of WMP sites originally numbering seven when first published (Elsasser and Contreras 1958) had just increased to thirty-seven.

Newly discovered WMP sites continue to be reported with no upper limit on the potential number of total sites currently in view. Just a few short months following the Ash Fork discovery, a new site was located in Flagstaff by a local rock art researcher, Robert Mark, just fifty miles east of Ash Fork along the old Atlantic and Pacific railroad corridor. Details of this site are briefly described in a post-script to this report. In addition, Summer 2021 brought word of a WMP site at Bishop CA originally located by rock climber, Charlie Harnack, and shared with the WMP research community by his climbing partner, Jerry

Oser. The Bishop WMP site sits at the base of the Sierra Nevada escarpment just north of a previously known site at Lone Pine. Both the Lone Pine and Bishop sites overlook the old Midland Trail from Salt Lake City, Utah (Marymor 2021b, pp. 4–6). Finally, a second WMP panel at Ash Fork missed on the initial surveys was reported in the summer of 2021 by a WMP enthusiast, Taylor Dearden (Figure 1). In mid-summer of 2022, the known corpus of WMP sites numbers 39 in all.[1]

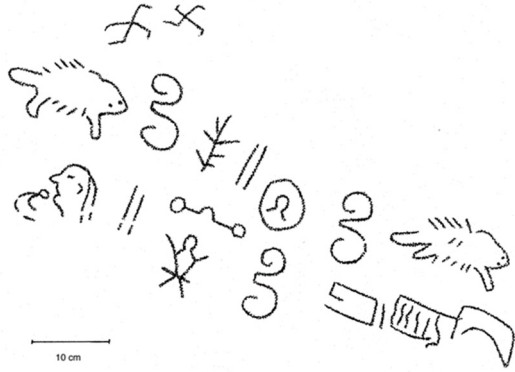

*Fortune flies away.*
*A diseased porcupine was treed. The man didn't know the porcupine was sick*
*when he ate of it. Death did not hunger that winter of the great snow.*

**Figure 1.** Western Message Petroglyphs, panel 2 at Ash Fork, Arizona (Drawing by David Lee)[2].

Dispersed across a wide region of the Western United States,[3] WMP sites are remarkably similar and recognizable based on their image content and organization, their style of execution, and by their geographic and historic landscape settings (Figure 2). Within the "classic" template for these sites variations can be recognized, most notably in California where the format shakes free of its classic structure at some site locations.

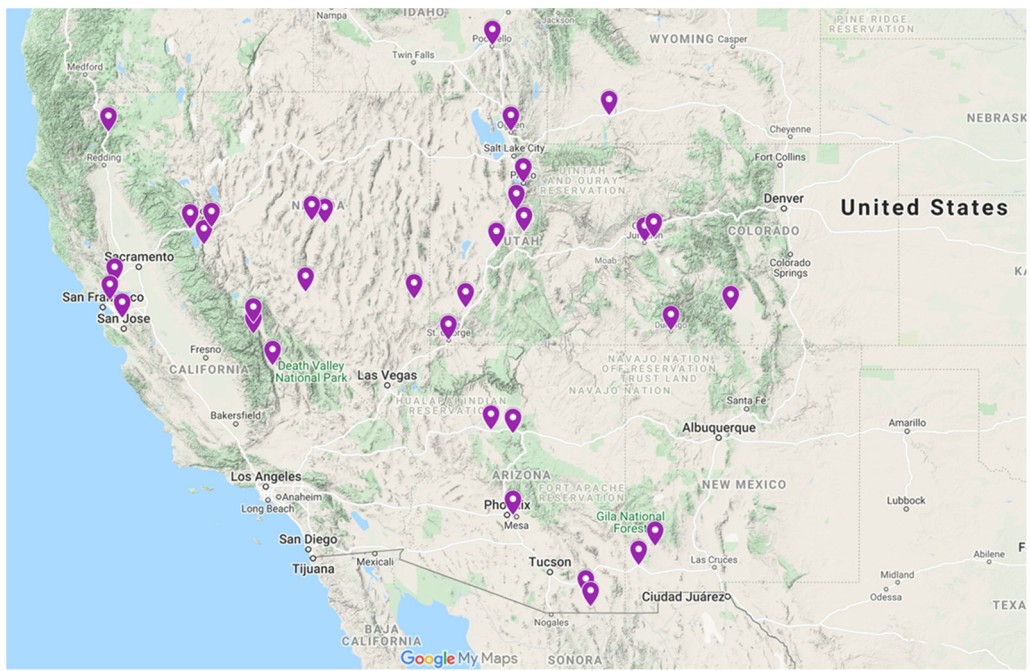

**Figure 2.** Western Message Petroglyph site distribution.

Western Message Petroglyphs are neatly engraved with metal edged tools into rock walls and isolated boulders located in natural landscape settings at 39 known sites (Table 1). WMP sites often include more than one engraved location with a total of 63 loci identified

so far. Tilden 1 in Berkeley, CA has the greatest number of loci, a total of seven boulders are scattered within a brushy scramble of one another on a steep canyon slope. WMP sites with two, three, and four engraved locations are common.

**Table 1.** WMP place names.

| Distribution of 39 Western Message Petroglyph Sites | | | | |
|---|---|---|---|---|
| AZ.1 | Ash Fork | | CO.3 | Durango |
| AZ.2 | Bisbee | | CO.4 | Grand Junction |
| AZ.3 | Flagstaff | | ID.1 | Pocatello |
| AZ.4 | Tempe | | NV.1 | Austin |
| AZ.5 | Tombstone | | NV.2 | Genoa |
| CA.1 | Berkeley-Tilden 1 | | NV.3 | Hickison Summit |
| CA.2 | Berkeley-Tilden 2 | | NV.4 | Pioche |
| CA.3 | Berkeley-Tilden 3 | | NV.5 | Tonopah |
| CA.4 | Berkeley-Tilden 4 | | NV.6 | Virginia City |
| CA.5 | Berkeley-Claremont Creek | | NM.1 | Lordsburg |
| CA.6 | Bishop-Birchim Canyon | | NM.2 | Silver City |
| CA.7 | Dunsmuir-Castle Crags | | UT.1 | Cedar City |
| CA.8 | Fremont-Mission Pass | | UT.2 | Fillmore |
| CA.9 | Fremont-Vargas Plateau | | UT.3 | Manti |
| CA.10 | Lone Pine-Alabama Hills | | UT.4 | Nephi |
| CA.11 | Rockville | | UT.5 | Ogden |
| CA.12 | Truckee | | UT.6 | Provo |
| CA.13 | Bishop | | UT.7 | St. George |
| CO.1 | Cameo | | WY.1 | Green River |
| CO.2 | Del Norte | | | |

　　Panels are defined by the planes and natural fractures of the rock surface that frame the spaces where the WMPs are engraved. A few panels contain more than one discrete composition of images. The classic WMP panel exhibits multiple images arranged in one row or as many as five parallel rows suggesting a text with semiotic rules (Figure 1). Often one or more "signifier" icons are set above, below, or to the side of the central text suggesting a header, footer, or perhaps a signature. In all, 77 engraved panels have been found among these sites, some with as few as one or two icons, and others with as many as 40 individual images. The Western Message picture-vocabulary includes images that appear only once, and many others that repeat within panels, and also recur time and again among the many sites. There is a total of 828 WMP images drawn from a base vocabulary of 214 distinctive picture-writing motifs engraved among the 39 sites (Figure 3).

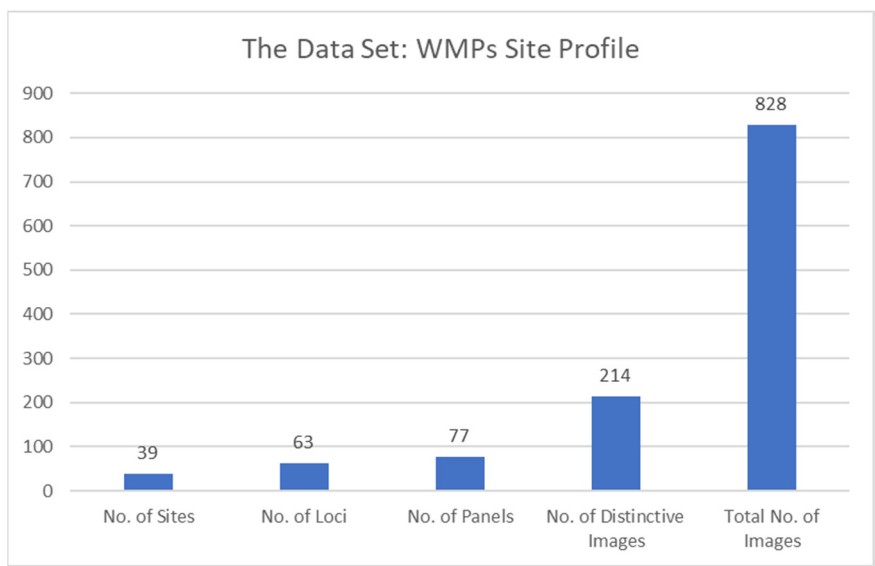

**Figure 3.** Western Message Petroglyphs site profile.

The symbol system is cobbled together into an original coherent whole from a variety of late historic forms of Native American picture-writing traditions, including: Ojibwa tree-bark drawings, Ojibwa Midéwiwin (Great Medicine Society) birch bark song-cycle notations, Dakota Sioux Winter Counts, Dakota Sioux Agency Roster picture-signatures, Shoshone petroglyphs, depictions of American Indian sign-gestures, and Navajo, Puebloan, Iroquois, and Southeastern Ceremonial Complex imagery—many of these tempered no doubt by the encroaching influence of Euro-American contact. The picture becomes more complicated as we take note that approximately 25% of the image content includes a bewildering variety of worldwide pan-cultural symbols, including: Egyptian, Maya, Chinese, Hindu, and possible Fraternal Order-inspired signs (Figure 4). The remaining 7% are from as-yet unidentified sources—some of these may be unique creations of the author(s).

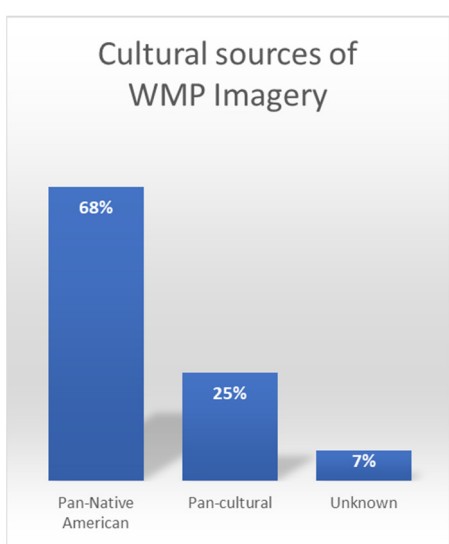

**Figure 4.** Cultural Sources of WMP Imagery.

In a previous WMP research paper, Marymor and Marymor (2016) analyzed repetitions of individual images, compounded images, and multi-image phrases that they observed among sites and demonstrated that all of the WMP sites share common authorship. The common author is likely to have been one individual, the central author, participating with a small group who were all "in-the-know".[4]

## 2. Geographic and Historic Landscapes

Across the West, the locations of Western Message Petroglyph sites cluster primarily in four regions: Far West, Great Basin, Southwest, and Rocky Mountain Region. The Google map shown here (Figure 5) illustrates the vast area where the WMP sites are found. An overlay of historic wagon roads and railroad rights-of-way reveals the narrow travel corridors that bind these sites together into a coherent whole.

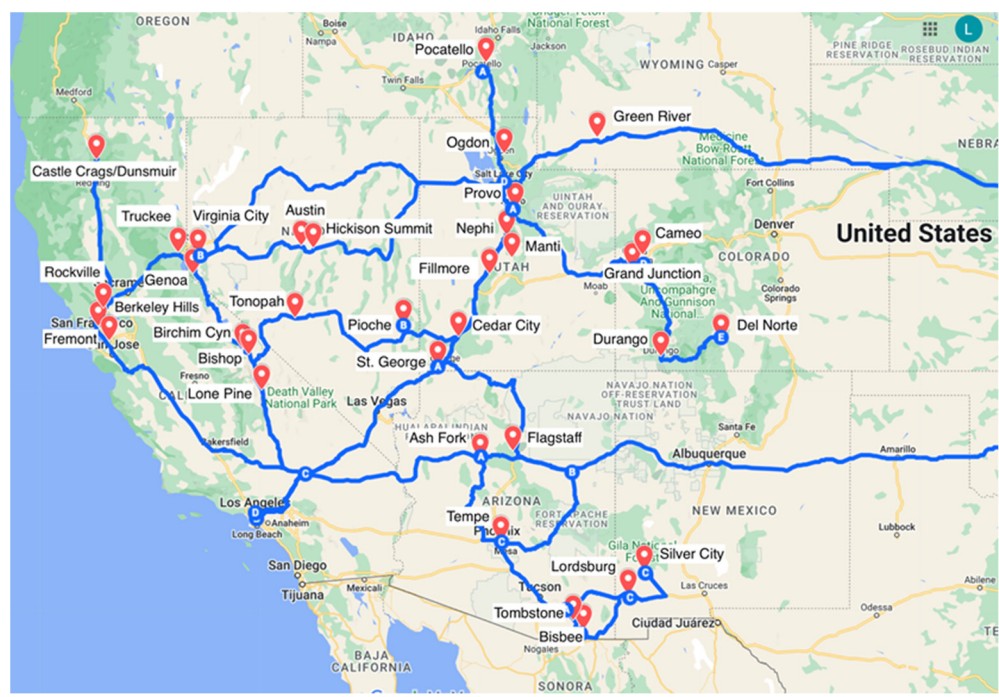

**Figure 5.** Routes of travel associated with Western Message Petroglyph sites.

A defining feature of WMP sites is that they tend to be located adjacent to, but somewhat remote from and overlooking historic routes of travel dating to the period of Western expansion (1846–1905). Positioned at particular nodes along these corridors, the sites are very often situated in relationship to town sites, and to mines and quarries that boomed and busted during this same period. Many of the sites map neatly onto a Mormon cultural landscape with over half of the sites overlooking Mormon towns and historic Mormon routes of travel, while a few are connected by virtue of Mormon commerce (Figure 6). These are primarily concentrated along the Mormon Corridor located along the western front of the Wasatch Mountains in Utah overlooking the Mormon Trail, the California Mormon Trail, and Mormon town sites. A second concentration of sites with Mormon associations is concentrated in the San Francisco Bay Area where sites cluster in the hills overlooking early Mormon settlements at Brooklyn Township (Oakland) and Washington Township (Fremont). A predictive model (See Marymor and Marymor 2016, p. 73) for where additional sites might be discovered anticipated subsequently reported sites in Tombstone, Arizona and in St. George, Utah. The model was based on a tri-fold criteria of proximity to historic routes, proximity to historic mines and/or quarries, and locations within the Mormon cultural region of the same era. Referencing the map in Figure 3 with the overlay of historic trails, rails, and WMP sites, it is easy to imagine fertile ground for future WMP site discoveries along these routes (See Appendix B).

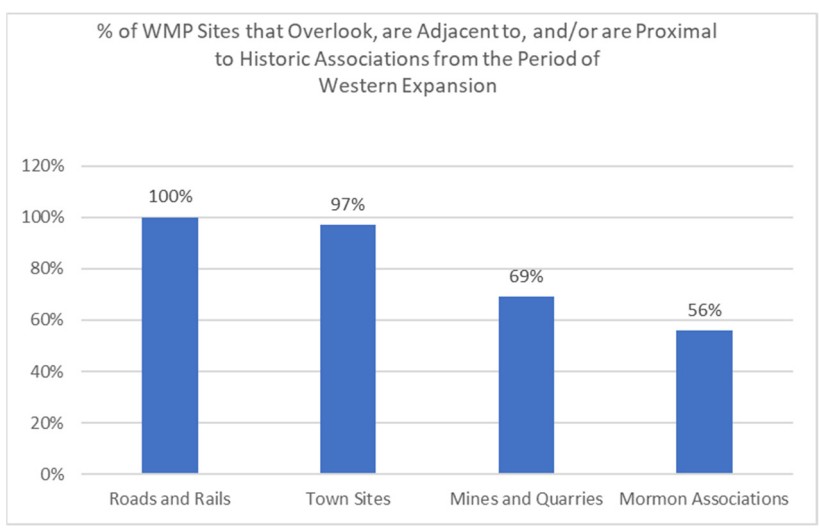

**Figure 6.** Western Message Petroglyph sites historic landscape contexts.

### 3. St. George, Utah, and Ash Fork, Tempe, and Tombstone, Arizona Exemplify How Geographic and Historic Landscapes Bracket WMPs in Time

*3.1. St. George, Utah*

The St. George, Utah Western Message Petroglyph site is a textbook example of the patterned convergence of geographic and historic landscape contexts in association with Mormon towns, quarries, and routes. The WMP panel (Figure 7) is engraved into a vertical sandstone face located at the base of the caprock with long views overlooking the town below (Figures 8 and 9). The original town site of St. George[5] was founded as a Mormon cotton farming mission in 1861 by Brigham Young, the President of the Church of Jesus Christ of the Latter-Day Saints. Young selected St. George to be the location of the denomination's first Utah Temple site. Construction was announced in 1871 with the Temple dedication following in 1877.

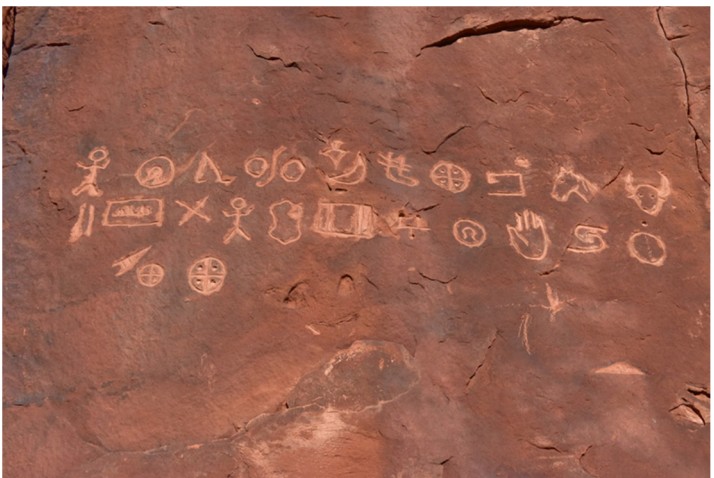

**Figure 7.** St. George UT WMP panel no. 1.

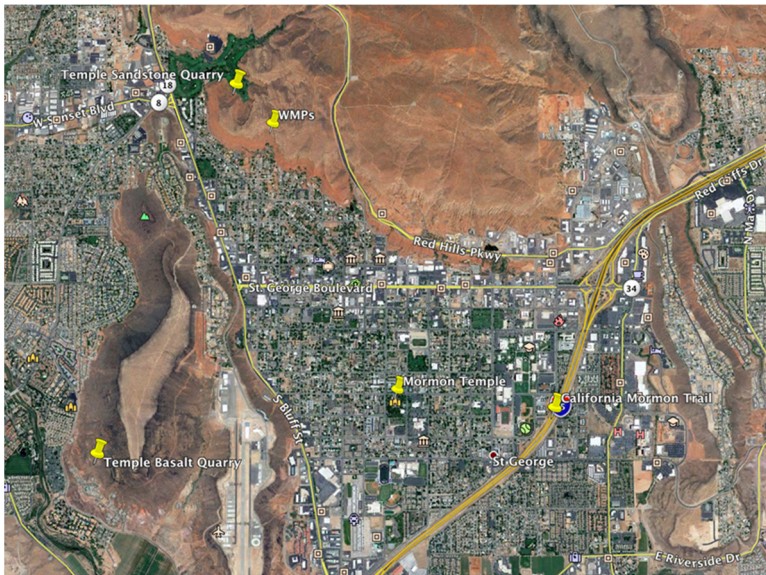

**Figure 8.** WMP site viewshed includes the Mormon Temple, temple sandstone quarry, and California Mormon Trail, with Temple basalt quarry nearby.

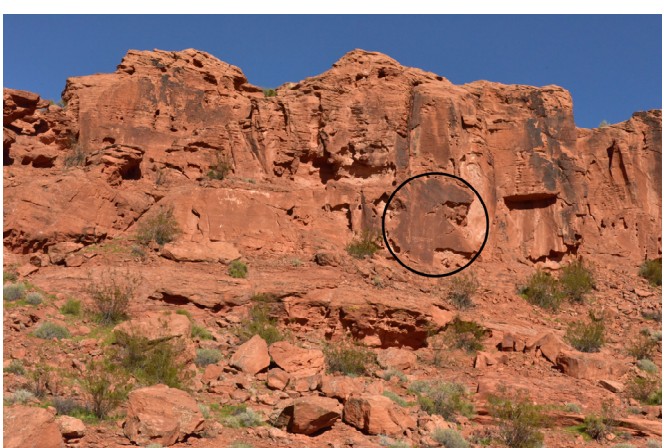

**Figure 9.** St. George WMP site location.

The black lava rock used in the construction of the Temple foundation and basement was quarried between 1871–1874 from a large volcanic formation located less than a mile away across the desert floor, and in line-of-sight from the WMP panel. The sandstone blocks that were used in the wall construction were quarried just a short distance away from the WMPs on the same landform where the WMP site is located. St. George, the southwestern most of the Utah WMP sites, is located along the California Mormon Trail whose route connected Salt Lake City with the church's distant settlement at San Bernardino, California. Like so many other sites strung out along the Mormon trails, the St. George WMPs look directly down at this historic trail which is roughly paralleled today by the route of Interstate 15 (Figure 8). St. George is also the terminus of the Mormon Wagon Road (later dubbed the "Honeymoon Trail"). Newlyweds living in remote Mormon settlements in Arizona traveled this network of arduous wagon roads to celebrate the sealing of their wedding vows with friends and family in ceremonies at the St. George Temple.

*3.2. Tempe, Arizona*

In Tempe, Arizona, two classic Western Message Petroglyph panels illustrate a second example where the geographic and historic contexts show a direct connection to the Mormon cultural sphere and routes of travel dating to the last quarter of the nineteenth

century (Figure 10). The placement of two adjacent panels located on Hayden Butte are found in the riparian zone set low on a vertical sandstone wall located along a dry wash that drains to the nearby Salt River. Although not unique among WMP sites, the riparian setting (Figures 11 and 12) is limited to a handful of examples[6] that contrast with the lofty overlooks the majority of sites occupy.

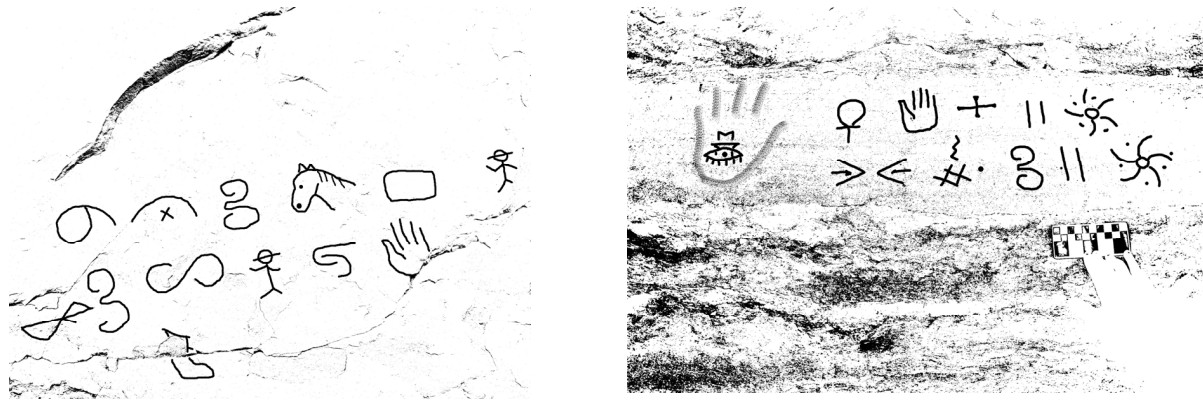

**Figure 10.** Tempe, Arizona. Panels 1 and 2. Digital photo enhancement with drawing on photo.

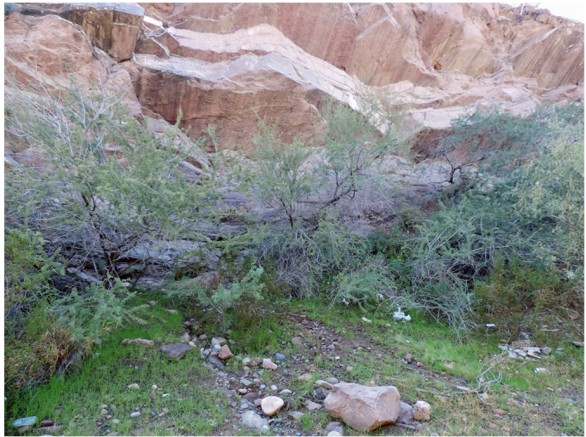

**Figure 11.** Tempe, Arizona WMP riparian setting.

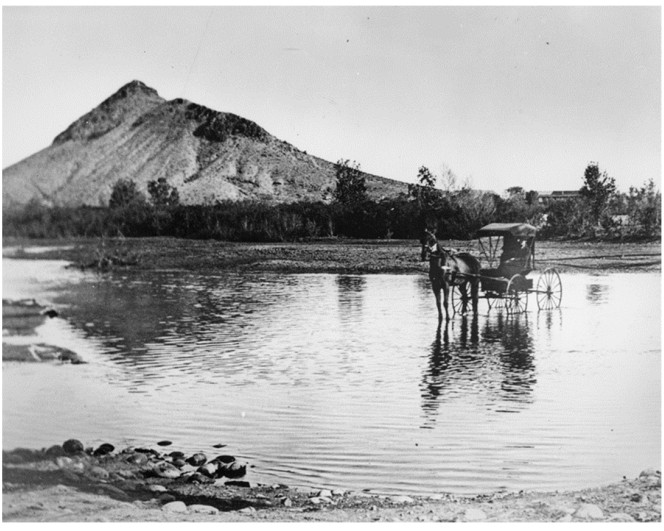

**Figure 12.** Salt River, Hayden Butte, Tempe, Arizona. Photographer unknown, 1870s–1880s.

The historic roots of Tempe, Arizona were established by Charles Trumbull Hayden, a merchant who moved into the Arizona Territory following the ratification of the Gadsden Purchase in 1854. Setting up shop in Tubac, and later Tucson, his stores served the local mining industry. In 1870 he purchased two sections of land at the north end of Hayden Butte for the purpose of establishing a mill, a general store, and a cable ferry. Beginning in 1877, it was Hayden who provided financial assistance and a welcome to the stream of Mormon settlers who were moving into the area to fulfill Brigham Young's vision for expanding Mormon settlements into Arizona Territory. The original destination for these settlers was the tiny U.S. Army outpost at Mesa (formerly Lehi). In 1882 Hayden was instrumental in advancing funds to Benjamin Johnson, with his extended family of seven wives and 45 children, who purchased town lots from Hayden to establish the first Mormon settlement at Tempe (Figure 13).

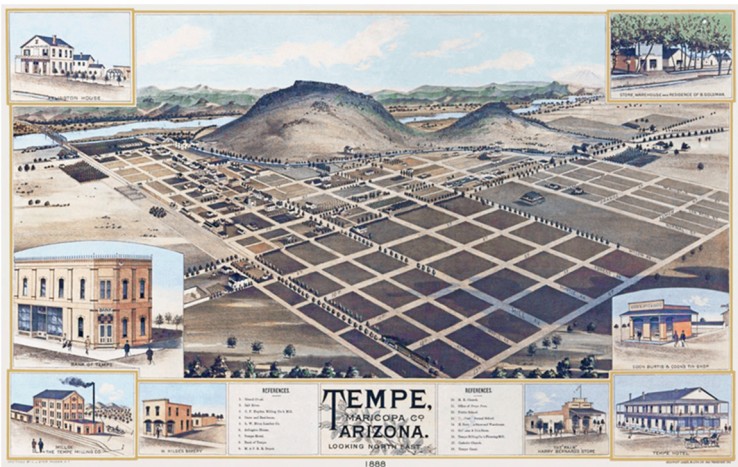

**Figure 13.** Tempe real estate boosterism map, 1888.

A likely route of travel for Mormons leaving Utah for the Arizona Territory in the late 1870s and 1880s would be along the Mormon Wagon Road, the aforementioned "Mormon Honeymoon Trail". This route opened Mormon access into the Arizona Territory beginning in 1873 and later provided return access for newlyweds from the Arizona settlements wishing to seal their vows at the newly constructed Temple in St. George. We can imagine the Johnson family leaving the Salt Lake area in 1882, traveling down the California Mormon Trail to St. George where they would skirt east along the Arizona Strip via Pipe Springs and Kanab in order to make the harrowing wagon crossing of the Colorado River at Lees Ferry. Having crossed the river, they would make their way south to cross the Little Colorado River near Winslow, traverse the White Mountain country in northeast Arizona, pass through the Mormon settlements at Joseph City, St. Johns, Snowflake, and Show Low, and finally arrive at Hayden Butte via the Salt River drainage (see map, Figure 5).

The WMP site at Hayden Butte is centrally located to this unfolding history. It is adjacent to the Mormon Wagon Road (Honeymoon Trail) at exactly the place where the historical figure in the person of Charles Trumbull Hayden welcomed and enabled Mormon settlement in the region.

The history of early railroad ventures throughout the West frequently intersects with the WMP story. Over two-thirds of the WMP sites overlook, are located adjacent to and/or are proximal to historic railroad rights-of-way. The WMP site at Hayden Butte exemplifies this pattern of association by proximity. In 1887 the Maricopa and Tempe Railroad bridge was constructed across the Salt River at the foot of Hayden Butte and by 1903 the Phoenix and Eastern Railroad tracks curved along the southern edge of the butte—both rail lines pass within view from the two WMP panels.

In most of these cases, WMPs are found at some distance from the tracks, in private settings, often with a view across the rail corridor from an elevated landform. We can

make inferences about the WMP author based on these patterned associations with historic landscapes that provide context for the WMP site locations. For the purposes of proposing a chronology for the sites and a theory of identity for the author this study assumes that the pattern of the sites' associations with historic rails (and trails, mines, quarries, and Mormon places) is not coincidental. Rather, it is proposed that the location of WMP sites indicates an intentional choice made by the author that suggests something of their personal history and interests. The patterned relationship between the engraving sites with historic landscapes guides us in the placement of the author in time. A temporal model based on the strength of the patterned site locations across the West suggests earliest probable dates for the sites. The model assumes that the episodes of engraving would not predate these historic associations (Table 2).

**Table 2.** WMP Historic context: Townships and railroad rights-of-way.

| Proposed window for engraving activity based on the historic context at three Arizona WMP sites | | | |
|---|---|---|---|
| Historic town established | | WMP Site location is adjacent to or overlooks railroad right-of-way | |
| | | Railroad arrives | Railroad |
| Ash Fork | 1882 | 1882 | Atlantic & Pacific RR (Santa Fe RR) builds the Ash Fork siding |
| | | 1895 | Santa Fe, Prescot & Phoenix Railway Phoenix to Ash Fork junction completed |
| Tempe | 1870 | 1887 | Maricopa & Tempe Railroad Bridge crosses Salt River at Hayden Butte |
| | | 1903 | Phoenix and Eastern Railroad tracks laid at Hayden Butte |
| Tombstone | 1877 | 1882 | Grading for the Fairbank to Tombstone right-of-way is begun by the Benson-Nogales Railway, but subsequently abandoned |
| | | 1903 | El Paso & Southwestern Fairbank to Tombstone branch line completed |

A summary of the site context of the Tempe WMPs highlights a cluster of dates within a narrow band all connected to the nexus of historic events associated with the location of the WMP site at the foot of Hayden Butte. These events include the arrival of Charles Trumbull Hayden in 1870 and the evolution of his ferry-crossing over the Salt River at the northern end of the butte into the small village of Tempe. Brigham Young's missionary aspiration that opened Mormon migration into eastern and central Arizona in 1877 was aided by Hayden's promotional aspiration for Tempe as witnessed by the loan and sale of Tempe townsite tracts to the Benjamin Johnson family in 1882. In the same years the Mormon Wagon Road passed through Tempe. The route guided Mormon missionary families traveling south from St. George into the region and shepherded Mormon newlyweds from the Arizona settlements northward in order to seal their vows at the St. George Temple. The subsequent growth of mining and agriculture in the region supported the coming of narrow-gauge rail with two lines passing within a "stone's throw" of the WMP site; beginning in 1887 with the construction of the Maricopa and Tempe RR bridge and the completion of the Phoenix and Eastern RR right-of-way in 1903.

From this history, it is reasonable to conclude that the earliest probable date for the Tempe WMP site occurs between 1870 to 1903. By tracing similar histories at other WMP sites, as explored through examples from Ash Fork to the north and Tombstone to the south, the proposal to set the historic floor for the WMP activity in the last quarter of the nineteenth century and opening years of the twentieth is strengthened. Following this,

a discussion of WMP imagery and identification of source material for specific images engraved at the WMP sites will further narrow the proposed time frame for WMP activity.

### 3.3. Ash Fork, Arizona

The town of Ash Fork, Arizona Territory, was established in 1882 as a small siding of the Atlantic and Pacific Railroad (a predecessor of the Santa Fe Railroad). Its *raison d'être* lay heir to the cascade of events set in motion in 1853 by the U.S. Congress' appropriation of funds and the appointment of Secretary of State Jefferson Davis to "Ascertain the Most Practicable and Economical Route for a Railroad from the Mississippi River to the Pacific Ocean.[7]" Of the two Southern Pacific surveys commissioned, Lieutenant Amiel W. Whipple's survey in 1853–1854 established a practicable route along the 35th parallel from Oklahoma to Los Angeles. Three years later the Pacific Wagon Road Office in the Department of the Interior assigned Lieutenant Edwin Fitzgerald Beale the task of building a wagon road across New Mexico and Arizona that would in large part follow Whipple's route along the 35th parallel. Beale accomplished the assignment with the aid of 100 road builders and his famous camel corps spanning the years of 1857–1860. Beale's Wagon Road was the first federally funded highway, subsequently replaced by US Route 66 and later by Interstate 40. The Beale Wagon Road passed 15 miles north of Ash Fork while Route 66 was routed through the center of town.

The importance of Ash Fork's location along the rail corridor increased with the completion of a north/south rail junction connecting Prescott and Phoenix with the east/west transcontinental line at Ash Fork. The Santa Fe, Prescott and Phoenix Railway (SFP&P) was completed in 1895. A spur line of the SFP&P located south of town connected Ash Fork with Jerome, the local gold, silver, and copper mining district (Figure 14).

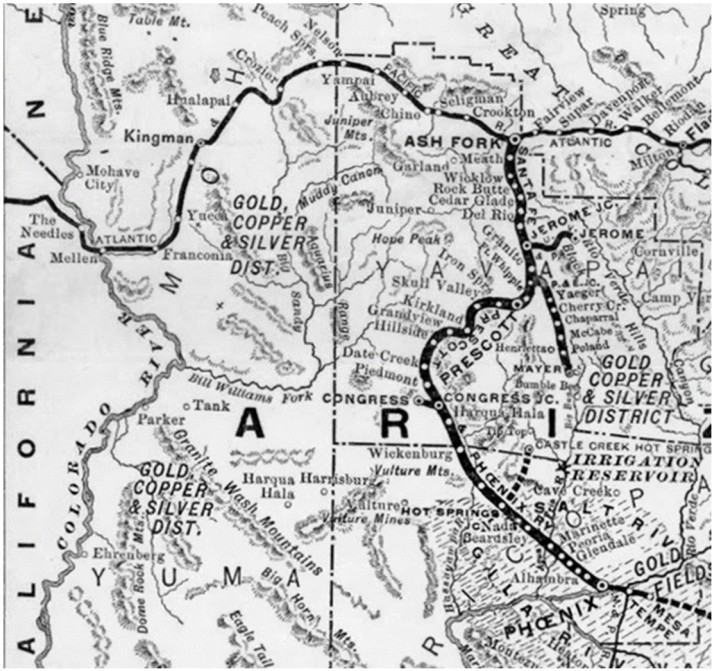

**Figure 14.** Ash Fork showing rail corridors (with transfer at Phoenix to Tempe).

The Ash Fork WMP site (Figures 1, 14 and 15) is engraved at the base of a low-lying lava flow situated south of town bordering the flood plain of Ash Fork Draw. The site is oriented toward the old town site and to the junction of the two historic railroad rights-of-way. The old SFP&P right-of-way passes 300' distant in front of the WMP panel. Following the completion of the SFP&P Railway in 1895, it became possible to make easy transit between Ash Fork and Phoenix, and continue on from Phoenix Station to Tempe via transfer to the local Maricopa and Tempe Railroad. The latter rail, as noted above passed

directly by the WMP panels at Hayden Butte, suggesting that this travel corridor may have connected the two sites. The earliest likely dates for the Ash Fork WMP site based on this historical context would be between 1882–1895 (the founding of Ash Fork and the completion of the SFP&P railroad).

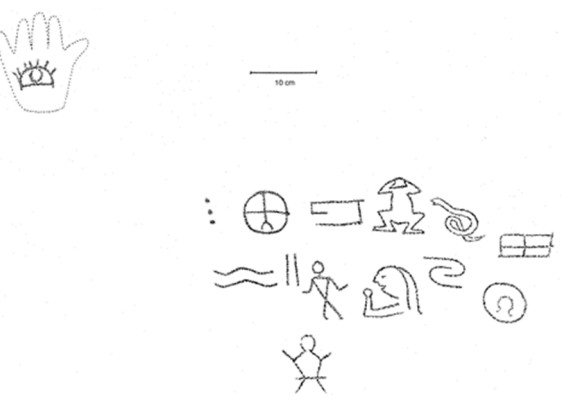

*The Witness.*
*For three years everywhere a great plague was cast down upon our fields*
*and rivers. When a person eats, unknown brothers go hungry.*

**Figure 15.** Ash Fork, Arizona, panel 1. Showing text with unique signifier to the upper left—a shadow of a handprint with radiant eye design inscribed in the palm (Drawing by David Lee).

### 3.4. Tombstone, Arizona

The Tombstone Western Message Petroglyph site (Figure 16) sits atop a modest knoll that rises above the surrounding desert flat with a sweeping view of the Tombstone townsite and mining district off in the distance.

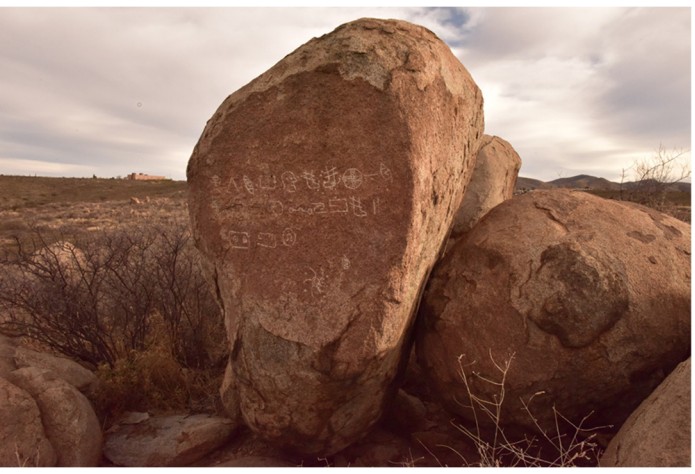

**Figure 16.** Tombstone, Arizona Western Message Petroglyph site.

The town and mining district boomed beginning in 1877 when prospector Ed Schieffelin discovered loose silver ore that had eroded out of a rich vein in the hills east of the would-be town. During the boom years Tombstone was the most prolific silver mining district in Arizona Territory (Figure 17). The end began to unfold in late 1881 when the ever deepening mine shafts struck ground water and one by one the mining concerns began to flood. By 1893 silver production in Tombstone was completely abandoned.

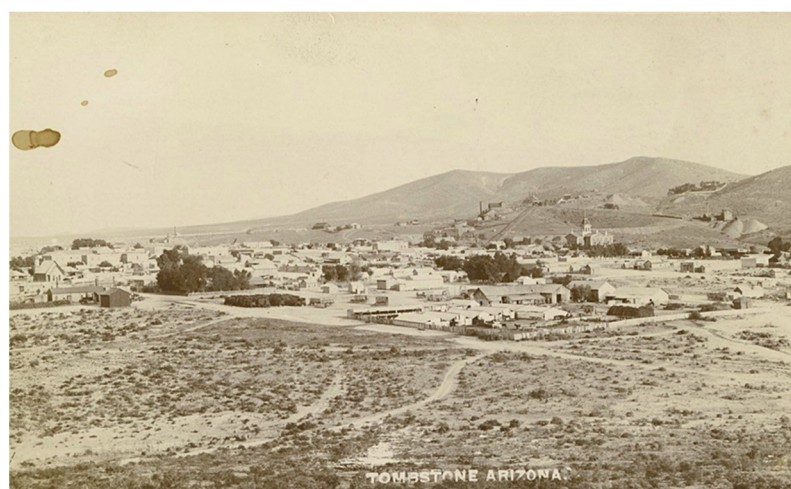

**Figure 17.** Tombstone, Arizona, 1880 (photographer unknown).

Of significance to the WMP story was the advent in 1900 of a second silver boom made possible when the Tombstone Consolidated Mines Company formed to take advantage of advances in steam pump technology that allowed for an economical method to drain the mines. Silver extraction subsequently resumed at depths below the water table. Tombstone Consolidated Mines Company bankrupted in 1911 following years of economic struggle with costly boiler and pump repairs, low ore values, and loss of production.

The Tombstone WMP site is accessed today from a narrow dirt road that winds its way out onto the desert flat from the edge of town. The track approaches the knoll with the WMP site and nearly brushes its flank before continuing on past toward an old crossing over the San Pedro River. Eroding in places out of the road bed one can today see the remnants of the narrow-gauge rails that were laid along this route in 1903 by the El Paso and Southwestern Railroad (EP & SW) to serve the needs of the reawakened Tombstone mining district. This short spur line connected Tombstone with Fairbank, a junction point for the main EP & SW line running from Benson to Nogales on the Mexican border (Figure 18).

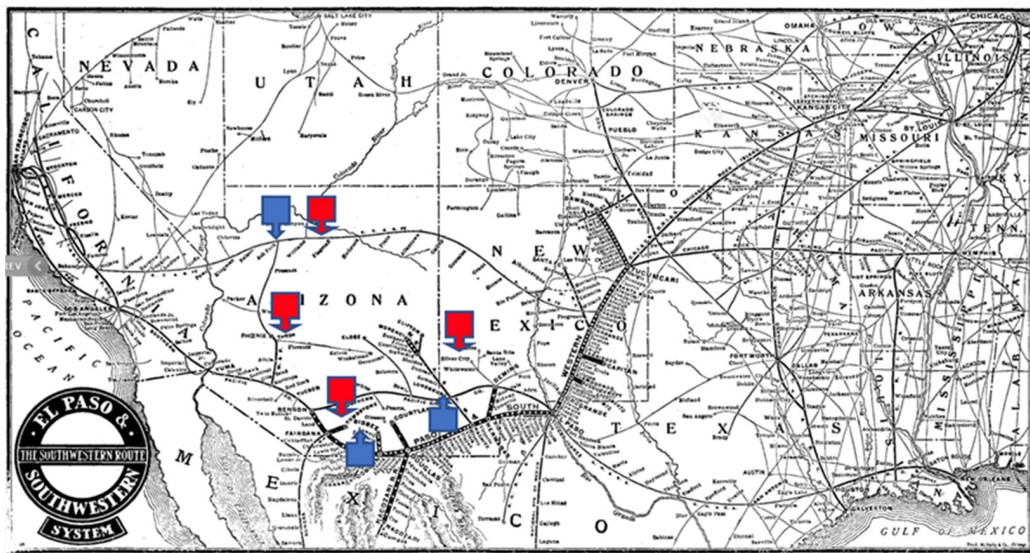

**Figure 18.** Red flags denote the distribution of the "pinwheel" motif along the southwestern RR corridors. Blue flags denote additional WMP sites located along these corridors.

The Tombstone WMP site shares the now familiar intersection between mining, town sites, rails, and the Mormon cultural sphere. The Mormon settlement of St. David lies twenty miles northwest of Tombstone along the EP&SW railway line. St. David was

founded in 1877 by Philemon C. Merrill, a former member of the Mormon Battalion, accompanied by a small group of settlers originally intending to settle at Mesa (Lehi), Arizona. The settlement began to thrive as a source of lumber harvested in the Huachuca Mountains and shipped via the EP & SW railroad to serve the mines and growing town of Tombstone.

The El Paso & Southwestern Railroad map (Figure 18) provides a useful canvas for illustrating the proximity of the seven southwestern WMP sites: Ash Fork, Flagstaff, Tempe, Tombstone and Bisbee, AZ, and Lordsburg and Silver City, NM. Strengthening the proposal that the path of travel between these sites is related to historic rail corridors is the observation that unique engraved icons are replicated among the sites along these routes, as well. A prime example is a circle form with radiating, serpentine arms—each arm intermingled with punctate dots. This image is only found at four WMP sites whose locations are flagged in red on the map. Unique WMP images that repeat among sites point

to the work of the same author moving among the sites.

The WMP engraving event is more securely connected to the second silver boom of 1900–1911 based on its location adjacent to the narrow-gauge right-of-way. During the early mining years of the 1870s–1880s the site location would have occupied a random spot on the desert floor. Prior to the discovery of silver in 1877 the whole of the San Pedro Valley, a region loosely "protected" by the Army outpost at Fort Huachuca, was a dangerous and lawless place dominated by Apache raiders and border-crossing desperadoes. Indeed, when Ed Schieffelin first made his way into the area to begin prospecting, soldiers at the fort advised him that the only stone he was likely to find would be his own head stone. Schieffelin's first mining claim was subsequently named "Tombstone".

## 4. Iconography

Analysis of the semiotic structure and image content of the 77 panels at the 39 Western Message Petroglyph sites has shed a considerable amount of light on this seemingly enigmatic symbol system. The classic structure of a WMP panel includes a central text consisting of one to as many as five lines arranged in parallel rows, often with one or more signifier images set above, below, or the to the side of the text. Other arrangements include stand-alone brief phrases of just a few icons, a clustering of icons with little or no linear structure, and one or two lone signifier icons with no accompanying phrase, text, or cluster (Figure 19).

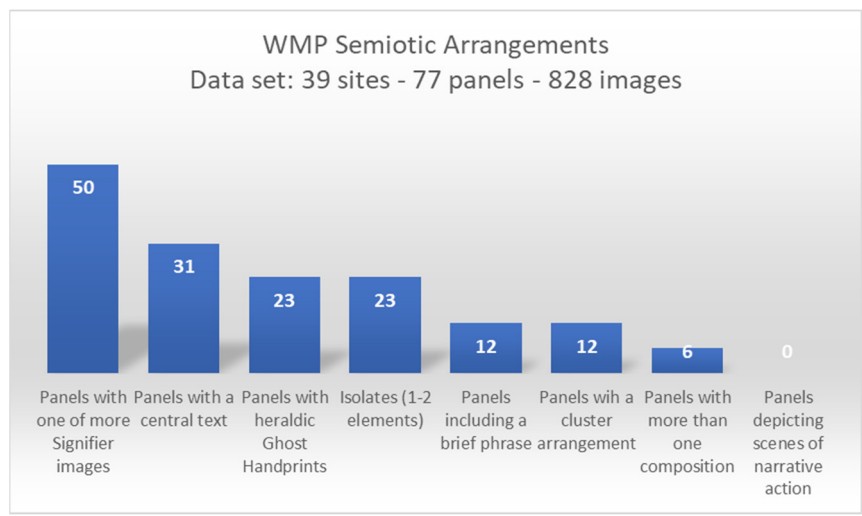

**Figure 19.** WMP semiotic arrangements.

Sixty-eight percent of WMP imagery is assembled from a diverse mash-up of proto-historic and historic Native American picture-writing traditions (Ojibwa, Sioux, Puebloan, Iroquois, Southeastern Ceremonial Complex, Shoshone, and American Indian sign-gesture language, etc.) intermixed with 25% pan-cultural icons (Egyptian, Maya, Chinese, Hindu, etc.). The remaining 7% of the icons originate from the author's unique inspiration or other sources as yet unidentified (Figure 4). The total image count for all of the panels across all 39 sites numbers is approximately 828 icons. These are drawn from a WMP lexicon of somewhat more than 214 individual images. As we shall see, the source for many of these images can be shown to originate in ethnographic publications of the day and reflect a "faux" picture-writing system of the WMP author's invention.

It is noteworthy that the symbol system is not an alphabet and the individual images do not spell out words. Each image performs the duty of an ideograph, and within the context of the overall text may function as a noun, verb, adjective, adverb, or a limited number of prepositions (before, after, above, below, etc.). Images can embed layers of meaning from the most representational, apparent interpretation to any number of synonyms within the bounds of the textual context as a whole (e.g., the icon for "snow" might also mean "winter" or "cold"). An individual image can embed much broader concepts in the same way that the "Red Cross" functions in modern western society to reference a culturally recognizable idea. Taken together, the whole of the text carries a messaging intent—one that can be approximated as we will see, but that would be difficult to verify without access to the author's contextual references.

In the examples where images are laid out in neat lines (e.g., Figure 16) we can see that the layout lends itself to reading left to right in accord with English, the Romance languages, and with an Ojibwa picture-writing tradition dating from the nineteenth century as noted in Hoffman (W.J. Hoffman 1888, pp. 216–17; reproduced in Mallery 1893). Ojibwa Midéwiwin scrolls were the property of a highly select initiated society of healers, also known as the Great Medicine Society. Their birch bark scrolls were closely held and brought out during secret ceremony where sequences of picture-icons assisted the chanter in recalling details of intricate song cycles. William J. Hoffman writing in 1888 described the exclusivity of the Midéwiwin scrolls and their picture symbols this way:

> "At Red Lake I discovered the existence of a secret chart which, according to the assurances of the Chief and assistant Medé priests, had never before been shown to a white man nor to any Indians, except those entitled to behold it or those who, after necessary preparation by preliminary fasting, were to receive instruction for the several degrees of the Midéwin". (Hoffman 1888, p. 217; See also Hoffman 1891; reproduced in Mallery 1893)

A symbol borrowed from Midéwiwin birch bark scrolls frequently appears in WMP texts in the form of two short, vertical, parallel line segments (||)[8]. The (||) notation signaled a pause between sections in the chanting. In WMP usage the symbol is employed as a grammatical device and functions as a comma or period. The preponderance of the (||) icons are found at the right-hand end of lines of text confirming that the WMP reading is meant to proceed from left to right (Figure 20). This grammatical device occurs more frequently than any other single image among the 39 WMP sites (n = 47) suggesting its importance in organizing the author's narrative phrasing.

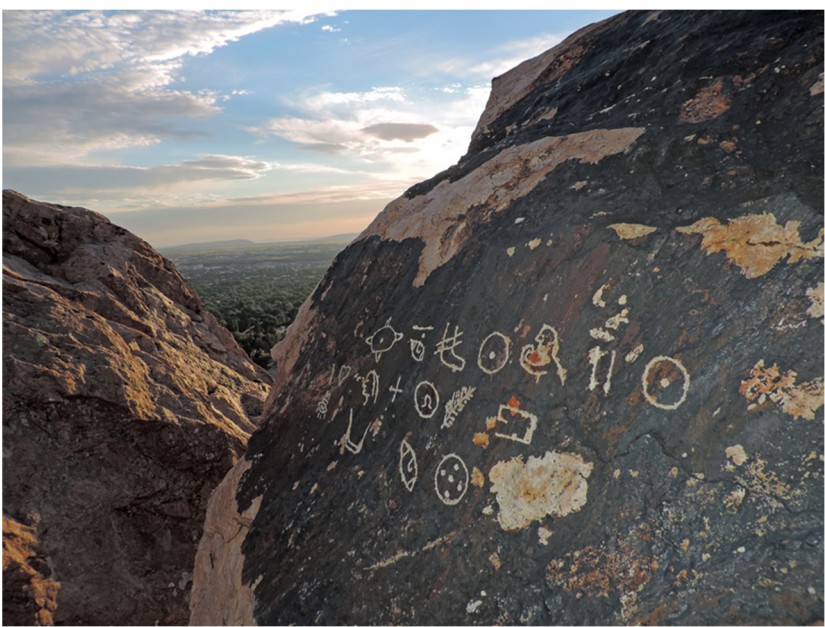

**Figure 20.** A classic Western Message Petroglyph text, Ogden, Utah.

Other semiotic conventions used in WMP texts include the symmetrical arrangements of strings of icons which may imply a poetic prose intent, the inversion of icons (an image can be flipped 180° to mean its opposite), and the repetition of icons (e.g., one teepee is a singular encampment; three teepees represent a village).

*4.1. A "Faux-Indian" Picture Writing Project: The Smoking Gun*

Between 1877 and 1893, Brevet Lieutenant Garrick Mallery, Acting Signal Officer of the United States Infantry, published a series of in-depth treatises on American Indian sign-gesture language and picture-writing in the U. S. Bulletin of the Geological and Geographical Survey of the Territories[9] and in the Annual Reports of the Bureau of Ethnology, Smithsonian Institution (Garrick Mallery 1880, 1881, 1886, 1893). Mallery's publications are well known to students of Native American rock art, especially his work appearing in the 4th Annual Report (Mallery 1886) and 10th Annual Report (Mallery 1893). He is frequently referenced as the first serious researcher of Native American rock art—his research culminating with the publication of the 10th Annual Report, a compilation of Native American symbolism from all parts of the United States along with in-depth comparisons to usages in other cultures from around the world (Figure 21). His essays on American Indian sign-gesture language that appeared in the Bureau of Ethnology Bulletin series (Mallery 1880) and in the 1st Annual Bureau of Ethnology Report (Mallery 1881) are lesser known, but are equally important to the Western Message Petroglyph story. Perhaps less well understood about Mallery's work is that his first interest was focused on American Indian sign-gesture language and secondarily with corresponding picture-images that he endeavored to show were related to it. He made a clear delineation between the symbolic and ceremonially inspired carvings and paintings found on rock surfaces in natural landscapes (rock art) and the American Indian picture-writing signs that were intended for more utilitarian communication following the format of sign-gesture language.

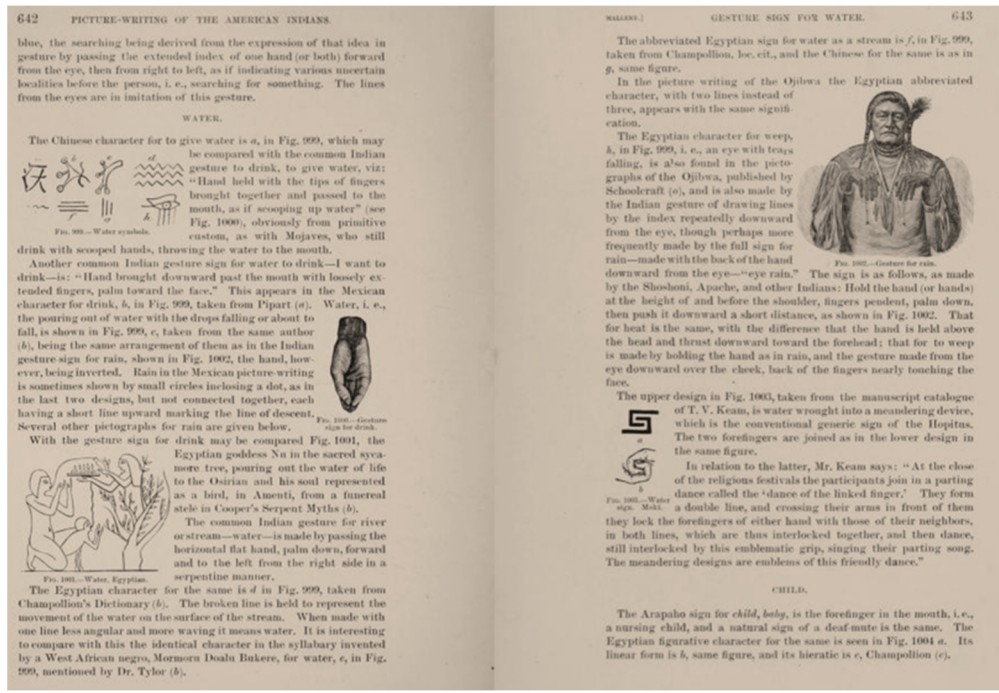

**Figure 21.** Garrick Mallery (1893). Tenth Annual Report to the Bureau of Ethnology.

In communication with colleagues and in searching the ethnographic publications of his day, Mallery was particularly pleased to discover likenesses from around the world among American Indian sign-gesture forms, American Indian picture-writing, and worldwide pan-cultural signs for similar concepts.

> The reproduction of gesture-lines in the pictographs made by our Indians seems to have been most frequent in the attempt to convey those subjective ideas which were beyond the range of an artistic skill limited to the direct representation of objects, so that the part of the pictographs, which is still the most difficult of interpretation, is precisely the one which the study of sign-language is likely to eludicate [sic: elucidate]. In this connection it may be mentioned that a most interesting result has been obtained in the tentative comparison so far made between the gesture-signs of our Indians and some of the characters in the Chinese, Assyrian, Mexican and Runic alphabets or syllabaries, and also with Egyptian Hieroglyphics. (Mallery 1880, p. 5)

Mallery observed that the use of American Indian sign-gesture language was mutually intelligible among the tribes roaming the Plains, but pointed out that dialects, vocabularies, and meanings of signs were regionally and culturally based (Mallery 1880, pp. 37–39).

### 4.2. Image and Author

Was the author of the WMP panels Native American, possibly of Metis lineage as suggested by some (Greer and Greer 2016), or an experienced Euro-American speaker of American Indian sign-gesture language who was conversant in American Indian picture-writing? The identity of the individual author is not known. However, authentic Native American picture-writing as found in the rock art record, on cultural artifacts, or as documented in the ethnographic literature simply does not exhibit the blend of multi-cultural imagery employed in the WMP panels. By combining pan-Native American picture-writing signs, graphic depictions of American Indian sign-gestures, and world pan-cultural icons the WMP author created a "faux Indian" expression. His selection of images mimics the broad ranging pan-Native American and pan-cultural content presented in Mallery's work. It is Mallery's publications that constitute our smoking gun for the source of WMP imagery (Figure 22). Of the WMP images that are not found in Mallery, many are found

in related ethnographic works of the day—especially those footnoted by Mallery in his various discussions of pan-cultural correlates with American Indian sign-gestures and picture-writing signs (For example: Jean-François Champollion (1836), George Copway (1850), W.J. Hoffman (1891), Lord Kingsborough, Edward King (Kingsborough, Edward King 1831–1848), P. Le Page Renouf (1875) and Henry R. Schoolcraft (1851) among others).[10]

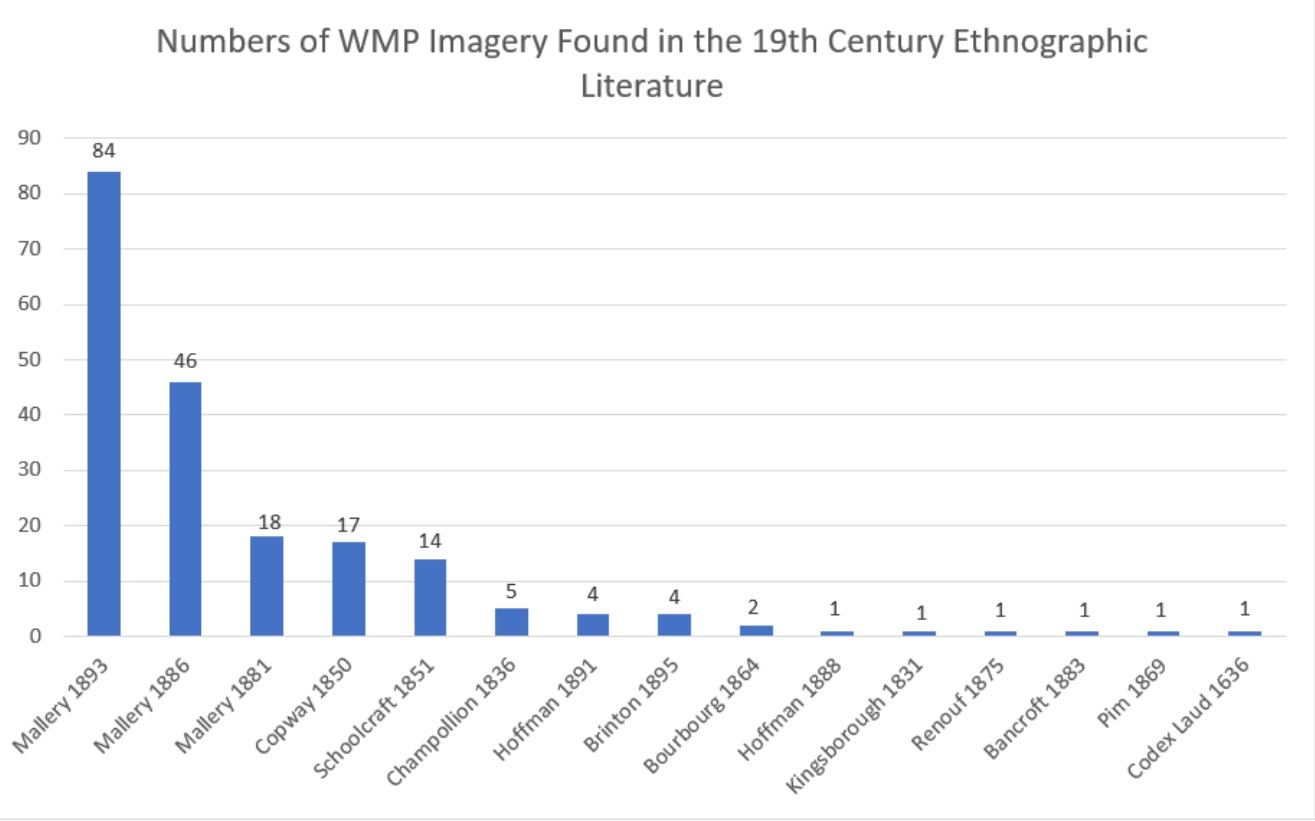

**Figure 22.** 19th Century sources for WMP imagery (Mallery 1881, 1886, 1893; Copway 1850; Schoolcraft 1851; Champollion 1836; Hoffman 1888, 1891; Brinton 1895; de Bourbourg and Éntienne 1864; Kingsborough, Edward King 1831–1848; Renouf 1875; Bancroft 1883; Pim and Seemann 1869; Codex Laud 1636).

In the study of WMPs, numerous examples have been identified where the author lifted unique images directly from the Mallery texts and placed them into the panels he engraved on the rock walls of the American West. While it is not certain if the WMP author had a pre-existing knowledge of American Indian sign-gesture language and picture-writing, his plagiarisms from Mallery, and other ethnographers and anthropologists of his day, make clear that his fluency with picture-writing was greatly enhanced by the encounter.

In each panel where borrowed images are found; the plagiarized images act as chronological stamps—the WMP panels can be no older than the lifted images' first appearance in their source texts. In most cases, this earliest date range aligns comfortably with the chronological indicators previously discussed relating to the WMP sites' historic landscape contexts. Certainly, many of the historic wagon roads and railroad rights-of-way adjacent to WMP sites would have originated as prehistoric Native American trails. However, the patterned placement of the sites at historic nodes along these routes is demonstrably tied to Euro-American developments resulting from the era of western expansion.

The proposal for the earliest probable date for each WMP panel can be narrowed by comparing the dates derived from the landscape contexts with the first publication date of the borrowed icons used in each panel. The Tombstone panel, for example, was assigned

an earliest probable date between 1900 and 1911 based on the site's association with the resurgence of mining and the construction of the adjacent El Paso and Southwestern Railway right-of-way. Noting the incorporation of the Ojibwa Midéwiwin ( | | ) sign for rest, or pause, that was first published by W.J. Hoffman in 1888, and again in 1891, one concludes that the panel could not have been engraved prior to 1888. Assuming that Mallery's publications were the WMP author's primary source, the best approximation of the earliest date would post-date the 10th Annual Bureau of Ethnology Report in 1893 where he featured Hoffman's work on the imagery of the Ojibwa Midéwiwin. Dismissing the remote possibility that the WMP author was an initiate of the Midéwiwin Society, it follows that the WMP author would not have learned the meaning and use of the ( | | ) icon prior to 1888–1893, and he would not likely have engraved it in the Tombstone landscape until after 1900.

With 56% of the WMP sites affiliated with historic Mormon associations, it is worthwhile to search for an author with connections to the Mormon cultural sphere. It can be deduced, having located our smoking gun in Mallery's work and in the publications that he carefully footnoted, that the author was well-educated and was writing in pictures by choice, rather than by necessity. In so doing, he displayed an intention to restrict access to the meaning imbedded in his texts. The remote access to the sites themselves and the quasi-philosophic and numinous tenor found in some of the texts suggests these writings may have been intended as private or closely held meditations. Further support for the argument for a Euro-American author can be found in the themes, storylines, "voice" and point-of-view expressed in the Western Messages themselves as developed in the following discussion on translation of WMP panels.

Three additional examples of images used in the WMP vocabulary illustrate the proposed smoking gun and are intended to establish the WMP author's direct connection to Garrick Mallery's publications.

### 4.2.1. "Old Man"

The description of an American Indian sign-gesture used to represent the attribute of "old man" appears in a discussion of the way an unintelligible sign-gesture became recognizable to Mallery. He gives the following account of a sign-gesture conversation with a Cheyenne Indian:

> A lesson was learned by the writer as to the abbreviation of signs, and the possibility of discovering the original meaning of those most obscure, from the attempts of a Cheyenne to convey the idea of old man. He held his right hand forward, bent at the elbow, fingers and thumb closed sidewise. This not conveying any sense he found a long stick, bent his back, and supported his frame in a tottering step by the stick held, as was before only imagined. There at once was decrepit age dependent on a staff. (Mallery 1880, p. 55)

A picture-writing correlate with the Cheyenne's sign-gesture for "old man" (Figure 23) is found in the 4th Annual Report (Mallery 1886) where it is illustrated as a personal name for one of many Dakota Indians registered in Red Cloud's band at the Dakota Agency in North Dakota (pl. LXI.59)[11]. A schematic representation of a human is shown in profile, bent at the waist, leaning forward and grasping a stick with a scrawled downturned crescent-shaped cloud over head. The name of the individual is given as "Old Cloud".

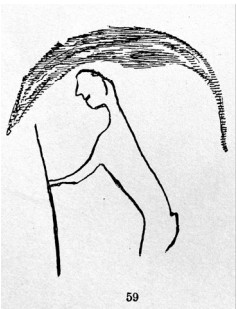

**Figure 23.** "Old Cloud" signature in Red Cloud's census, Pine Ridge Agency, Dakota Territory, 1884 (Publ. in Mallery 1886, 1893).

Two additional depictions of the "old man" sign are found among the American Indian picture-writing examples assembled by William Tomkins (Figures 24 and 25). These are included in his 1926 monograph, *Universal Indian Sign Language*, and in its 1929 revision, which he developed for use by the Boy Scouts of America. His handbook became a popular publication issued in souvenir edition for the Boy Scout World Jamboree held in England in 1929. As a child from 1884–1894 Tomkins "lived at the edge of the Sioux Indian Reservation in Dakota Territory, located at Fort Sully, Cheyenne Agency, Pierre . . . "[12]. There he worked on the range, befriended local Indians, and began his study of Sioux language and signs. His knowledge of American Indian sign-gesture language was actively pursued first-hand with numerous native speakers among the Blackfoot, Cheyenne, Sioux, Arapahoe, and others. Tomkins' monograph is primarily taken up with an illustrated compendium of American Indian sign-gestures, but includes a substantial glossary of picture signs which he assembled primarily from Mallery's 10th Annual Report. He also referenced Henry Schoolcraft's publications on Native Americans in the 1850s, and other historic ethnographic accounts. His depiction of "old" (Tomkins 1926, p. 74) presents an "old man" in Euro-American dress (Figure 24). In a later edition (Tomkins 1929, p. 77) Tomkins added a second image representing the depiction of old that closely resembles Old Cloud's signature in Red Cloud's Dakota Agency Roster (Figure 25). We find in Tomkins' glossary numerous examples of picture-writing that do not appear in Mallery, or any other familiar source—some may come from sources known only to himself, and others may be adaptations, modern, or of his own invention although he asserts all are genuine Native American signs (Tomkins 1926, p. 67).

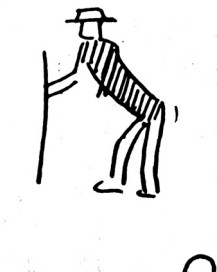

Old

**Figure 24.** Picture-writing image for "old" reimagined by Tomkins for his Boy Scout monograph (Publ. in Tomkins 1926).

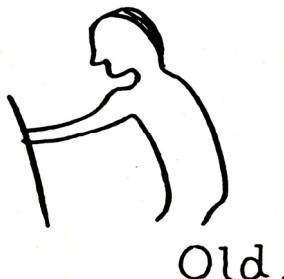

**Figure 25.** Picture-writing image informed by Old Cloud's signature in Mallery (1886) (Publ. in Tomkins 1929).

The WMP author, too, uses some signs that to-date have no known source. These may represent prior experience with picture-writing that underlies his study of Mallery, or they may represent his unique adaptations and inventiveness. In his essays, Mallery stressed the importance of flexibility and adaptation as key attributes required of American Indian sign-gesture speakers and by extension, picture-writers (Mallery 1880, pp. 37–39).

4.2.2. "None, Nothing, Negation"

In both the 1st (1881) and 10th (1893) Annual Report, Mallery engaged in a discussion of the American Indian sign-gesture for the concept of "none, nothing, or negation" and two pan-cultural picture-writing traditions similar to it. He specifically remarked on the similarity of the formal shape of the American Indian sign-gesture to both Egyptian and Maya hieroglyphs. These similarities evoked a sense of Mallery's wonder at the universality of the sign-gesture form, the similar formal characteristics of the pan-cultural correlates, and the fact that in all these cases the gesture and images are interpreted as meaning the same: none, nothing, or negation.

> The sign [-gesture] for none, nothing, sometimes used for simple negation, is made by throwing both hands outward from the breast toward their respective sides.

> With these compare the two forms of the Egyptian character for no, negation, the two upper characters in Figure 1006 taken from Champollion. No vivid fancy is needed to see the hands indicated at the extremities of arms extended symmetrically from the body on each side.

> Additionally, compare the Maya character for the same idea of negation, the lowest character of Figure 1006, found in Landa . . . . (Mallery 1881, p. 356; 1893, p. 645)

The WMP author stylized a unique compound image of the "old man" sign at Tonopah and Virginia City, Nevada with all of the required formal characteristics present: human in profile, bent at the waist, leaning forward, grasping a walking stick with out-stretched arm (Figure 26a–c). The Egyptian hieroglyphic form for negation (Figure 26d) floating above creates a compound glyph transliterated as "not–old man" and can be translated as "young man".

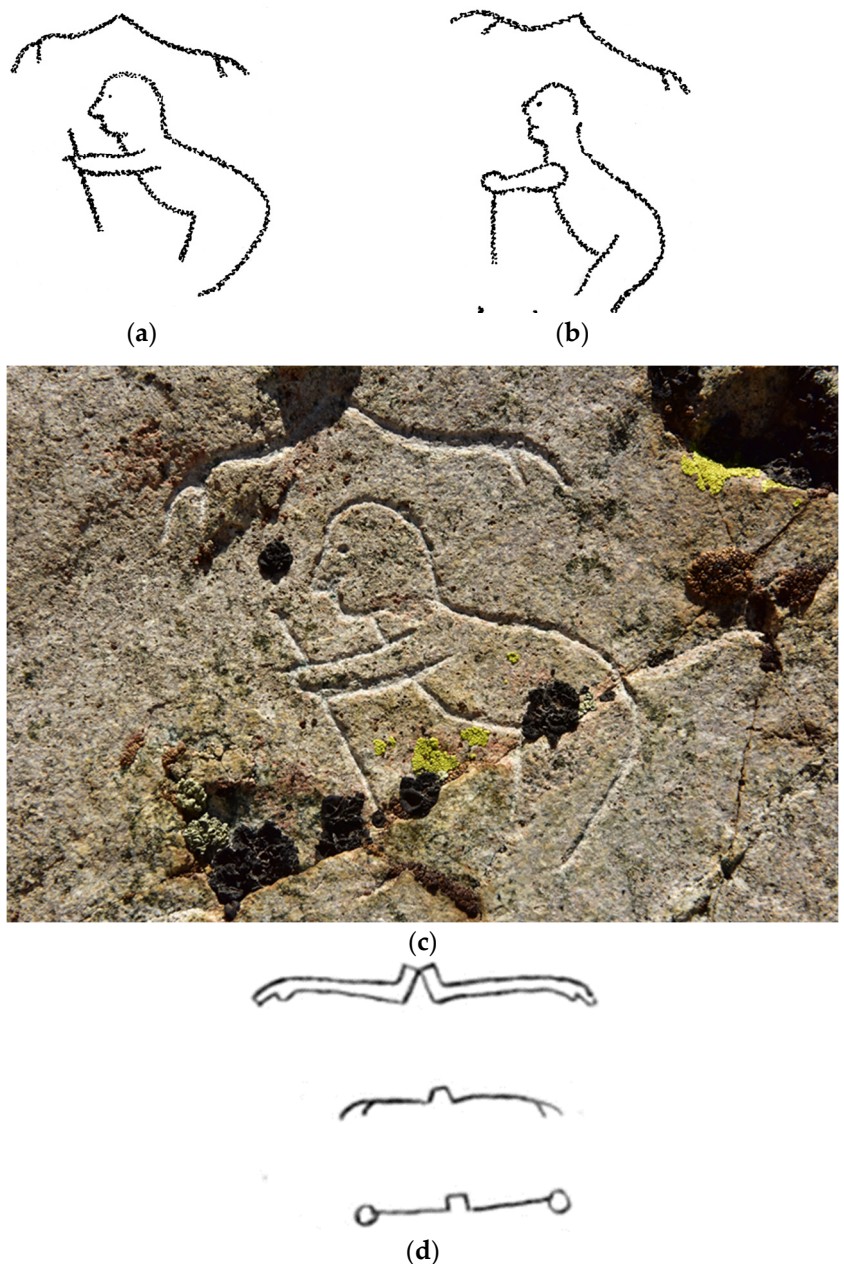

**Figure 26.** (**a**) "Not old man" (e.g., young man). Western Message Petroglyph (Virginia City, NV). (**b**) "Not old man" (e.g., young man). Western Message Petroglyph (Tonopah, NV). (**c**) Detail. Virginia City, NV: "not-old man", e.g., young man. (**d**) Egyptian and Maya signs for negation. Mallery (1881, 1893).

The convention of compounding the old man sign with a second one floating above follows Mallery's (1880, p. 62) discussion on the grammar of American Indian sign-gesture language and picture-writing. It conforms neatly with both the compound format of "Old Cloud's" signature in Red Cloud's Dakota Agency roster, and with Jean-François Champollion's (1836) *Grammaire Égyptienne* where he illustrated this "down-turned arms" shape floating over a second icon and interpreted it to indicate "none of" the entity below it. Mallery carefully referenced Champollion, the French decipherer of the Rosetta Stone, and pointed his reader to his 1836 publication. There are many other uses of both the Egyptian and Maya forms of the negation sign in WMPs both as a compound glyph floating above, or sometimes as an individual glyph conditioning the sign that comes before or after it. In

his use of these compounded images the WMP author is caught in the act of plagiarizing directly from the 10th and possibly from the 1st Annual Report to the Bureau of Ethnology.

4.2.3. "Weeping-Eye"

Perhaps the most iconic image in the Western Message Petroglyph picture vocabulary is the "weeping-eye" motif (Figure 27) which Marymor and Marymor (2016) formerly referred to as the "all-seeing eye". In Western Message form this eccentric engraving is rendered as a weeping eye surmounted by a pedestal and the letter M. The incorporation of the letter M individualizes the use of the "weeping-eye" in its Western Message context. The icon is found at 11 WMP sites in its classic form, and at five additional sites in variant forms (See Section 8. Postscript). Typically, the image is employed as a "signifier" set above, below, or to the side of the main body of a linear text. In some cases, the image has been engraved into the palm of a red painted handprint—the pigment long weathered away has left only the engraved "weeping-eye" visible to today's site visitors.

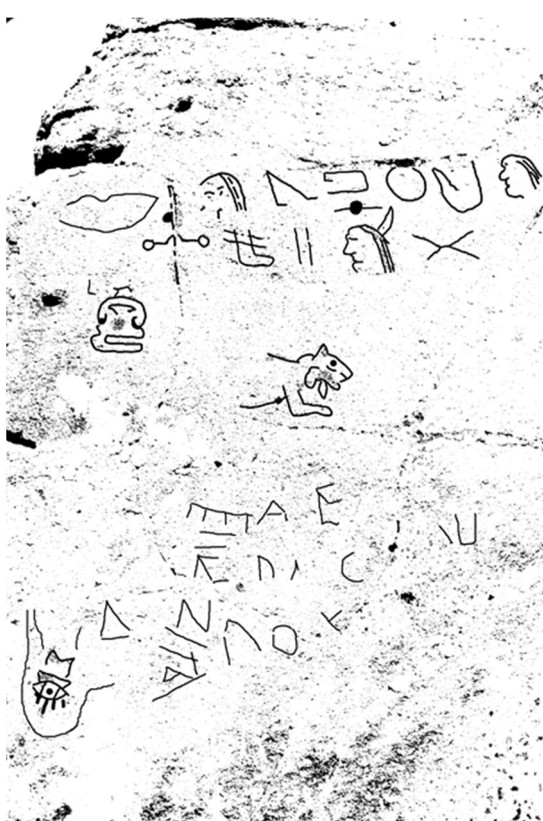

**Figure 27.** Del Norte, Colorado. The Maya negation sign appears in the first position of line 2. The weeping-eye is inscribed in a "Ghost Handprint". Digital tracing from photo.

Examples of a weeping or radiant eye motif are found in a wide variety of cultural contexts—the Egyptian hieroglyphic rendering and the radiant eye captured within the apex of a pyramid on the back of the U.S. dollar bill are immediately familiar to most. An eye depicted within the palm of a forward-facing handprint is familiar in a number of Fraternal Order contexts and appears in prehistory in Aztec, Maya, Southeastern Ceremonial Complex[13], and Kwakiutl contexts (Rands 1957).

To locate a source for the WMP rendering of the weeping-eye image we need stray no further afield than Garrick Mallery's text which, as demonstrated, is central to the WMP author's project. In a discussion in the 1st Annual Report Mallery begins with a description of the sign-gesture for "rain" (Figure 28a):

The sign for rain, made by the Shoshoni, Apache, and other Indians, is by holding the hand (or hands) at the height of and before the shoulder, fingers pendent, palm down, then pushing it downward a short distance, Figure 114 . . . (Mallery 1881, p. 116). . . . The Egyptian character for weep [Figure 28b], an eye, with tears falling, is also found in the pictographs of the Ojibwa[14], and is also made by the Indian gesture of drawing lines by the index repeatedly downward from the eye, though perhaps more frequently made by the full sign for rain, described on page 344, made with the back of the hand downward from the eye—"eye rain"[15]. (Mallery 1881, p. 143)

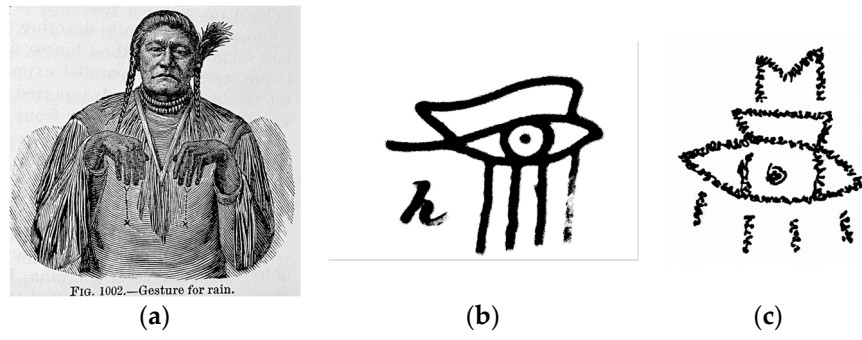

**Figure 28.** (**a**) Indian sign gesture for "rain". (Mallery 1881, Figure 114; 1893, Figure 1002). (**b**) Egyptian character for "weep". (Mallery 1881, Figure 142; 1893, Figure 999.h; after Champollion 1836). (**c**) Western Message Petroglyph picture-writing image (Ogden UT; typical).

Mallery's depiction of the Egyptian hieroglyphic correlate with an American Indian sign-gesture is a prime example that exposes the WMP author's propensity to lift images from Mallery's text and make them his own. The weeping-eye from the Ogden, Utah site (Figure 28c) illustrates the WMP author's individuation of the motif.

## 5. Faux Indian Picture-Writing in Historic Context

To place the Western Message Petroglyph author's project in some further historic context, it is important to note that he was not the first, nor the last person to turn to the nineteenth century ethnographic literature with the intent to employ the imagery in "faux Indian" picture-writing pursuits. Garrick Mallery concludes his narrative in the 10th Annual Bureau of Ethnology Report with a caution to be on watch for fraudulent examples of American Indian picture-writing and offers an example co-opted from George Copway's (1850) publication (Figure 29) among others. William Tomkins' Boy Scout handbook first published in 1926 is an example of a modern attempt to repurpose Native American picture-writing imagery mined from Mallery's work and place it into the service of young boys' romantic Native American pursuits of honor and merit (Figure 30).

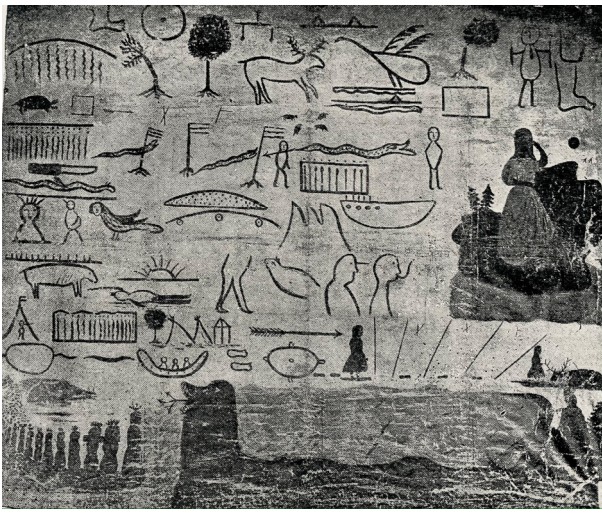

**Figure 29.** From Mallery (1893, Figure 1289). Fraudulent Ojibwa picture-writing.

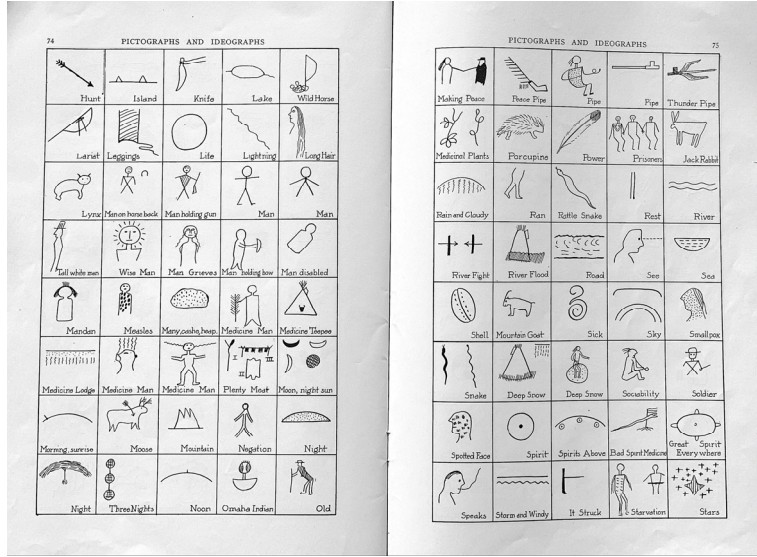

**Figure 30.** Illustration from *Universal Indian Sign Language* (Tomkins 1929).

## 6. Transliteration and Translation: "We Do Not Know When Death Goes Hunting"

Garrick Mallery's discussions concerning grammar and syntax are informed by his years-long study of American Indian sign-gesture language. According to Mallery, the method of reading late historic picture-writing can be derived from the conventions pertaining to American Indian sign-gesture language (Mallery 1880, p. 4). In an effort to read the Western Message Petroglyph texts Mallery's observations on rules of grammar and syntax are applied to the WMP panels.

- Icons represent thought concepts related to the images' formal characteristics and those meanings can be extended by applying useful synonyms (Mallery 1880, p. 56).
- The same sign can represent diverse meanings and concepts (Mallery 1880, p. 58).
- Generally speaking the signs are objective and rarely have a mystical intent (Mallery 1880, pp. 53–54).
- A sign whose meaning is known can be taken to mean its opposite if shown in inverted, or flipped, orientations (Mallery 1880, p. 62).
- Grammatical devices can be recognized, specifically the Ojibwa sign for pause, or rest (|  |) when used grammatically acts as a comma, or period (Hoffman 1888).

- Adjectives most often follow the noun that they modify, in the Spanish fashion [e.g., casa blanca; house white] (Mallery 1880, p. 63).
- Signs represent root concepts and can be compounded [e.g., great spirit + house = great spirit's house = church] (Mallery 1880, p. 62).
- Signs are radicals; the basic part of a word to which other parts may be added [e.g., hunt, hunting] (Mallery 1880, p. 56).
- Sentence structure is not rigid. Generally read in an orderly sequence left to right and top to bottom, as presented, but extracting its meaning is often derived by allowing the images to fall into a sensible [gestalt] (Mallery 1880, p. 58).

Translation begins with a transliteration for each picture sign whose definitions are sourced from the nineteenth century ethnographers where we have them, and from twentieth century interpreters to fill in gaps (e.g., William Tomkins 1926, 1929; Ernest Seton 1918; Robert Hofsinde 1959; Tehanetorens 1998). In some cases, there are no known historic definitions to turn to at all. In these cases, the translator is confronted with "white spaces" in the text. If the white space is too broad, an impasse develops and prevents a coherent translation. Comparing a sign with unknown meaning in context with other panels where it appears allows the testing of tentative translations. In some cases, poetic license allows the translator to arrive at an approximation. In all cases, the best translations can be no better than the attempt to recover the WMP author's intent tempered by the translator's own preconceptions. Proceeding with the assumption that the WMP author is attempting to communicate a message, the attempt is made to receive it by grounding oneself as best as possible in what is known about this historic era of American Indian sign-gesture language and picture-writing.

The spreadsheet in Appendix A illustrates the historic sources and the definitions for each picture-image found therein. These definitions are applied to each image in sequence as they occur in the Green River, Wyoming panel no. 2. The result is an image-by-image, line-by-line transliteration of the text. The translation moves beyond the "word salad" that results from the orderly transliteration. The attempt at translation is more of a poetic exercise than a scientific, falsifiable one. Referencing the Wyoming Green River panel (Figure 31a,b), the results of this method of transliteration are presented, followed by the proposed translation which allows a plausible meaning to emerge, one that is restrained by the historic definitions of the signs. The two "signifiers" are shown in parentheses.

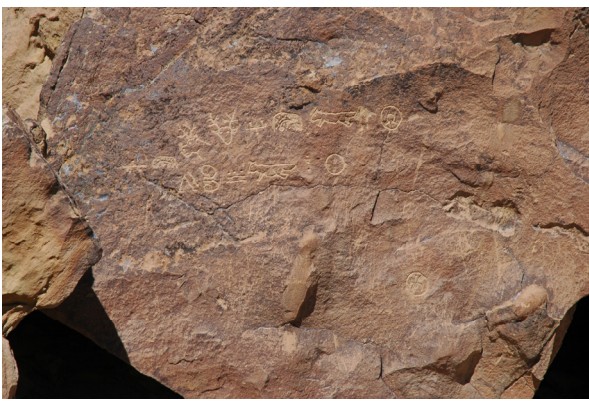

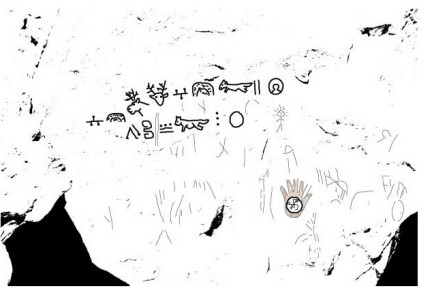

The Transliteration:
Bad-winter-Elk-Deer-good-winter-Fox-pause-hidden/obscure–
motion/hunt-death-pause-three/road-Fox-three-life.
(Wise Man).
(Incised Ghost Hand Signifier).

The Translation:.
*"A bad winter for Elk and Deer is a good winter for Fox. We don't know when death goes hunting. Fox pursues three roads, each of the three a life.*
*(Wise Man).*
*(Incised Hand Signifier).*

(**a**)       (**b**)

**Figure 31.** (**a**) Green River, Wyoming. Panel 2. (**b**) Green Riverm Wyoming. Panel 2 rendered to black and white, with tracing on photo. Transliteration of the imags from ethnographic sources, followed by proposed translation.

### 7. Summary: "You Did So, You Did So" (Figure 32)

The wide distribution of Western Message Petroglyph sites throughout the western United States with their patterned associations to historic landscapes including trails, rails, town sites, mines, quarries, and Mormon places, and their formulaic structure and shared imagery suggest that these picture-writing sites are bound up with the history of the settlement of the American West during the last quarter of the nineteenth century and opening years of the twentieth. Their typical settings in remote and elevated locations that overlook the adjacent historic landscape, and their use of a mix of pan-Native American and pan-cultural icons suggest a restricted or private intent in their messaging. Whether created by a sole author, or a small group of individuals who were all in-the-know, it is evident that the central actor to this unfolding mystery was well educated with access to the ethnographic literature of his day. It is in these publications that he located many of the diverse culture-bound images that he repurposed for use in the WMP project.

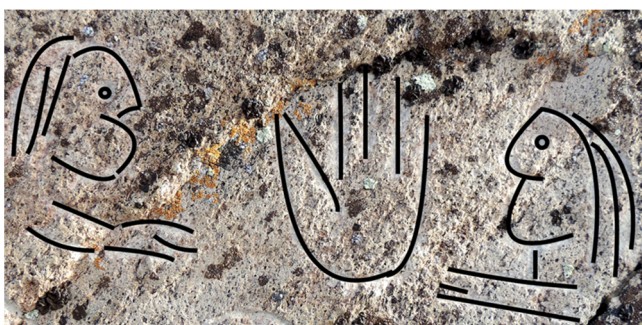

**Figure 32.** Tonopah, NV detail, "You"—did so—"You". (Digital tracing).

The identity of the central author remains a mystery, but it would seem productive to search among individuals with roots in the Church of Jesus Christ of the Latter-Day Saints (LDS) based on the high percentage of sites associated with Mormon routes, places, and commerce. The histories of mining and railway expansion in the nineteenth century West are bound together and the WMP author's connection to either, or both, is evidenced by the high percentage of sites that are associated with these—so perhaps searching for a mining or railroad man with a skilled hand for neat engraving will prove successful. An educated individual who elected to communicate in picture-writing by choice, rather than by necessity suggests an intention to restrict access to the meanings imbedded in his panels. A valuing of restricted access to knowledge may have been acquired from membership in a Fraternal Order, in the LDS Church, or both. Finally, the over-arching themes, transliterations, and attempted translations reveal a meditative, philosophic, quasi-numinous tenor to the messages themselves. The content of the WMP texts appears to be written in a poetic-prose voice and often expresses sympathy for, or a wry understanding of the human condition. The WMP project ultimately sheds light on a quirky undercurrent of western settlement as yet missing from the histories of western expansion. Our author appears to be a true American original.

### 8. Postscript: "Ghost Hands" and the "Weeping-Eye"

*8.1. The Flagstaff Western Message Petroglyph Site*

On 31 August 2020, barely four months after the prior discovery of a WMP site at Ash Fork, AZ, news arrived from Robert Mark, a rock art researcher who resides in Flagstaff, Arizona, that he had just located yet another Western Message Petroglyph site while volunteering on an archaeological survey for the City of Flagstaff. The Flagstaff WMP site is located in a jumble of freestanding basalt boulders just below the rim of the low-lying, north–south trending McMillan Mesa that bisects the city. Adjacent boulders give way to a small, semi-secluded open space in front of the engraved rock which stands approximately 5.0′ high by 8.0′ long. In days past, as in the present day, this secluded space would have

provided privacy and some measure of protection from the elements. A small perennial rill runs just below the site. All taken together, the site makes a fine location as a temporary encampment, and is currently used in this way by modern transients, as evidenced on the day of our visit by the presence of cast-off wine bottles and an abandoned suitcase. Looking south from the mesa top above the WMP site is an unobstructed view of the historic Atlantic–Pacific (aka Santa Fe) railroad right-of-way and the adjacent route of the historic Beale Wagon Road.

*8.2. "Ghost" Handprints*

In addition to neatly fitting the classic mold for WMP sites in regard to landscape and historic contexts, the Flagstaff site adds to a growing number of variant WMP sites. At Flagstaff we find the remnants of two lone signifier glyphs engraved on the vertical face of the host boulder without any accompanying picture-writing text (Figures 33 and 34). Both of the signifier images are common to the WMP lexicon—the pinwheel motif is briefly discussed in the body of the text above (see p. 14) while the handprints and weeping-eye motif are discussed at some length below.

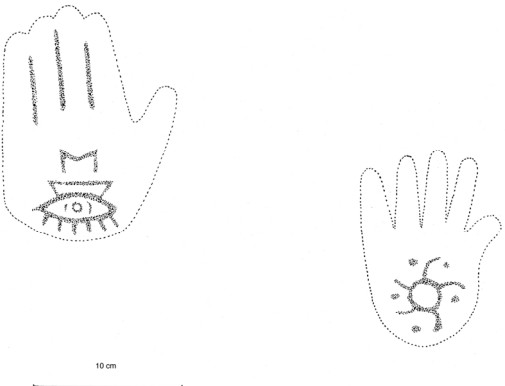

10 cm

**Figure 33.** Flagstaff WMP panel. (David Lee drawing).

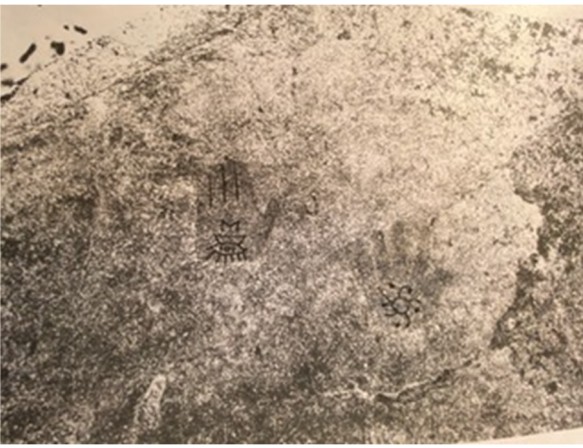

**Figure 34.** Digital photo enhancement using PhotoShop ®. Drawing on the resulting image illustrates an approximation of the faded handprints.

Digital photo enhancement has recovered evidence of two individuals' handprints that had been coated with pigment and impressed upon the rock—each hand engraved with a familiar WMP design motif. In all cases where we have WMP sites with remnant painted hands, it is always the left hand that was impressed upon the rock. It is probable that the result intended was to confront the viewer with a semblance of a right hand with an embellished palm (rather than the left hand facing away from the viewer while showing

a design on the back of the hand). The composition of a painted hand with an engraved palm design is a heraldic sign by its nature—forward facing, static, and steeped in symbolic content.

Historic photographs from three WMP sites (Berkeley, Castle Crags, and Fillmore) dating from the 1940s and 1950s depict similar handprints. At the time these images were taken a thick red paste, perhaps prepared from hematite or ochre, was readily observed as the colorant. At the twelve WMP sites where evidence of 23 faded heraldic prints has been recovered (Figure 35) we find the residue of pigment has completely weathered away. Only a ghostly image and a few lightly incised outlines remain as evidence of the hands for the discerning eye aided by computer enhancement. However, the palm engravings are still readily visible in most cases, and it may well be that some signifier glyphs at other sites were once set in the palms of handprints that we are no longer able to see.

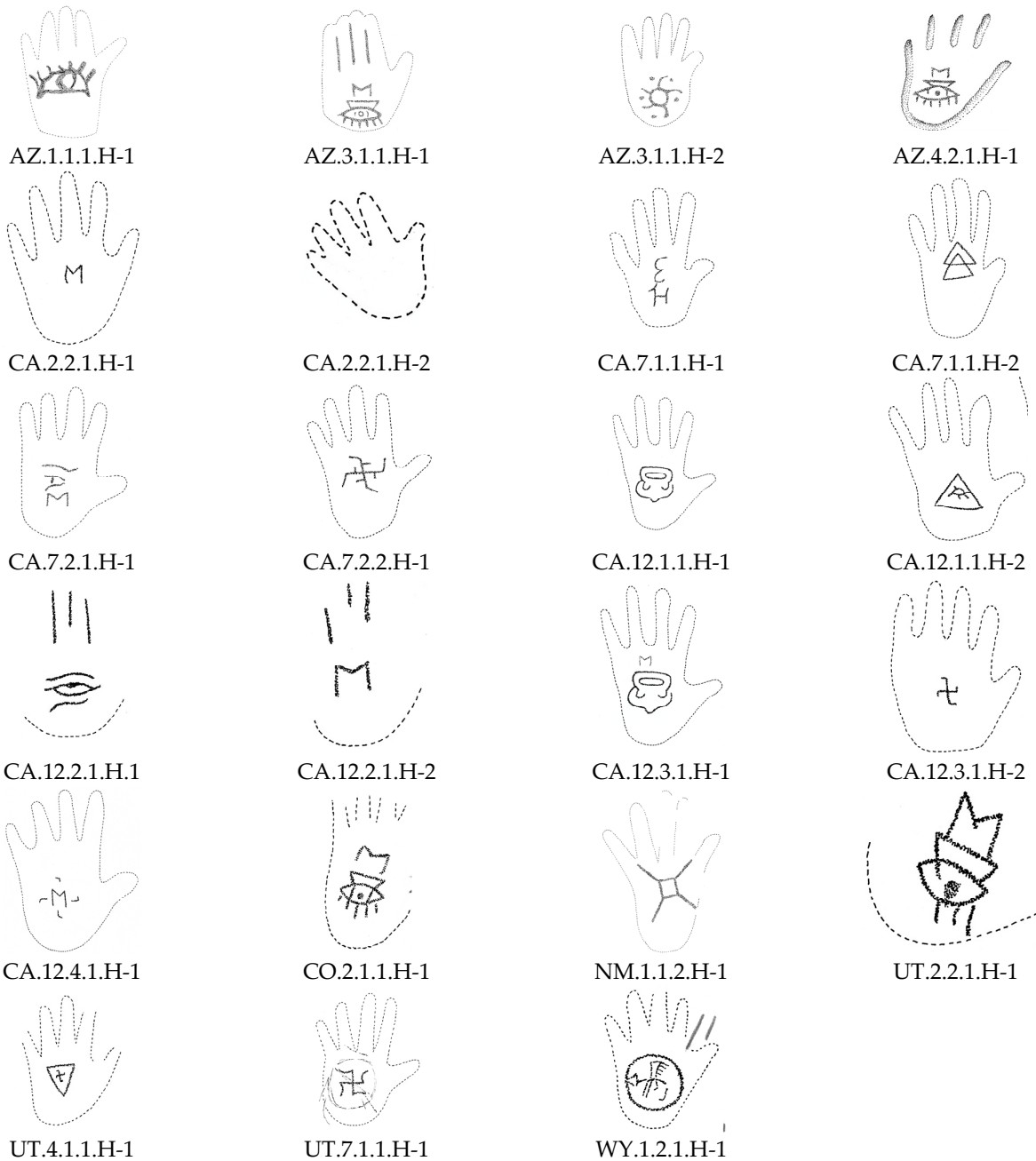

**Figure 35.** "Ghost" handprints (drawings by David Lee).

The weathering rate of the painted handprints provides a rough measure for estimating the relative age of WMP sites. For example, a newspaper report published in the Berkeley Daily Gazette (Hansen 1953) includes a photograph of two painted handprints at the Tilden 2 site (Figure 36). The archaeologists who reported and published this site, Al Elsasser and E. Contreras of UC Berkeley, described what they took to be red hematite pigment coloring the hand impressions. When shown the site thirty years later by Elsasser, this author was unable to see any remnant of the pigment whatsoever. Thus, a handprint that was impressed upon the rock likely no earlier than 1893 was completely weathered away by 1983—the pigment persisted for less than ninety years. Only an engraved letter M that had been engraved in the palm of the larger of the two prints remains as evidence of the original composition.

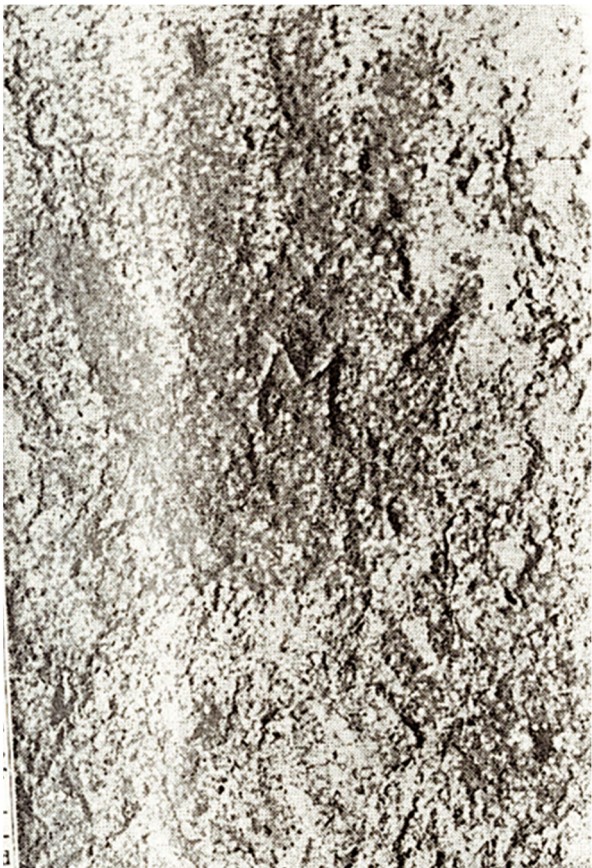

**Figure 36.** Tilden 2, locus 2. Berkeley Daily Gazette (Hansen 1953).

Pairs of handprints are found at four WMP sites (Flagstaff, Truckee, Berkeley and Castle Crags). We can observe signs of individuality within the pairs of handprints based on the sizes of the prints and the difference in the shape of the thumbs—one straight, and the other bent, or recurved.

Bent thumbs (Figures 34 and 37), which result from a recessive gene, are present at several of the WMP sites (Flagstaff, St. George, Truckee, and Castle Crags). None of the hands at the four sites that include hands with bent thumbs carry the same engraved palm design. It would be difficult to make the argument that all bent thumb handprints represent the same individual, but it would be premature to dismiss the possibility. Examples where heraldic prints with the same palm design are repeated among sites suggest the movement of the same individual from place to place. The weeping-eye is a good example to illustrate this (See Section 8.3).

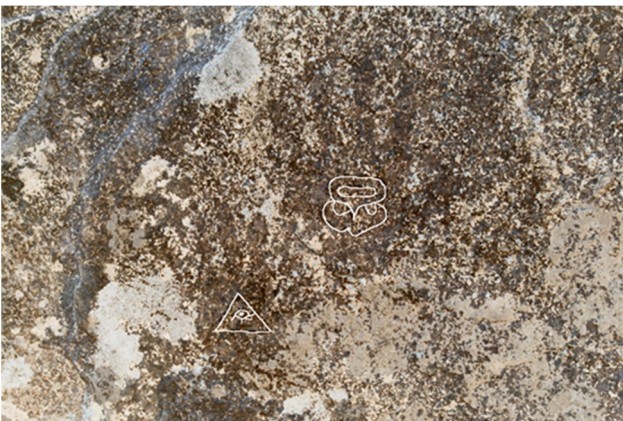

**Figure 37.** Truckee CA, Locus 3. Photo digital enhancement of one of three pairs of handprints found at this site. Notice the bent thumb on the upper hand. A lone seventh print appears on an isolated boulder at Locus 4.

The upper handprint at Flagstaff incorporates a classic weeping eye with pedestal and letter M executed with sharp edges (Figures 33 and 34). Three vertical lines above the motif appear to delineate the middle three fingers of the hand print. This convention is repeated at Truckee CA, where an Egyptian style weeping-eye appears with three vertical lines above, also delineating a barely perceptible hand print (Figure 38). This Truckee hand is paired with a second hand. In this case, the hand has an engraved letter M in the palm of a barely perceptible hand print, also delineated by three incised vertical "finger lines" above. The stylistic convention of engraving three vertical lines to highlight the middle fingers at Flagstaff helps us feel more secure in asserting that the Truckee motifs where the ghost shadows are much more difficult to see were also once set in painted hands.

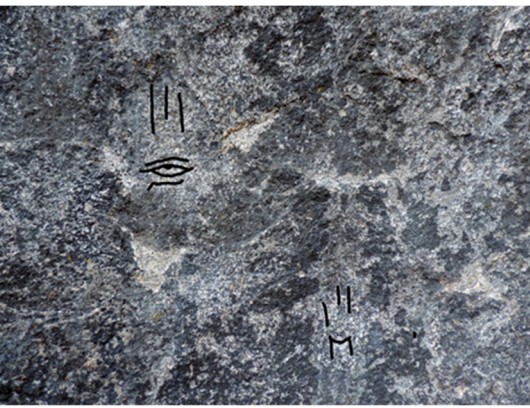

**Figure 38.** Truckee CA, locus 2 (enhanced with drawing on the print to make the design elements more visible).

### 8.3. Distribution of the "Weeping-Eye" Image

The Egyptian weeping-eye motif at Flagstaff appears in its classic form (borrowed from Mallery 1881, p. 358, Figure 142) at a total of 11 WMP sites and at five additional sites in variant forms. Seven of the 16 appear as palm designs (Figure 39) while the remainder appear as stand-alone signifiers. The map (Figure 40) illustrates the distribution of sixteen weeping-eye and radiant-eye motifs among the 39 WMP sites. The illustration helps visualize the shared authorship of WMP sites based on these repeated designs that overlay the routes and historic places that interconnect them.

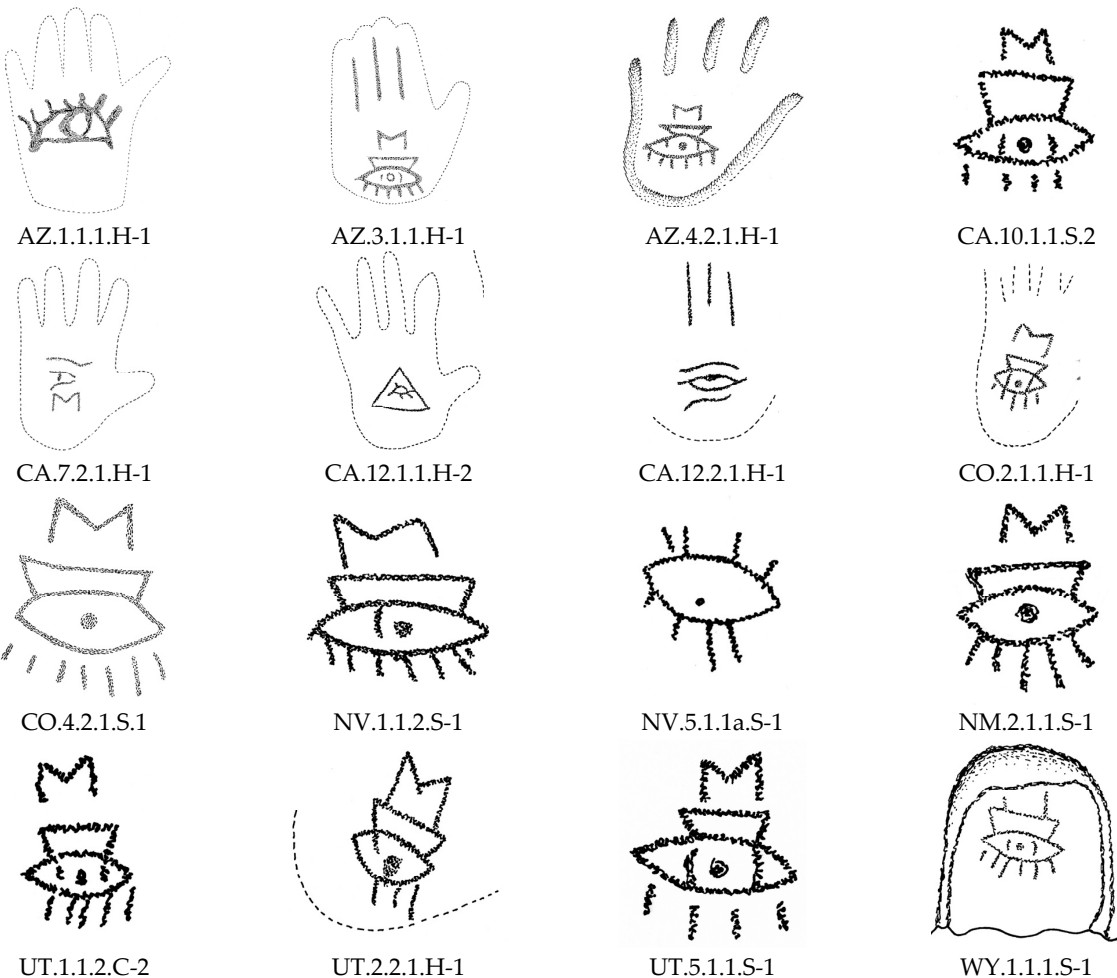

AZ.1.1.1.H-1     AZ.3.1.1.H-1     AZ.4.2.1.H-1     CA.10.1.1.S.2

CA.7.2.1.H-1     CA.12.1.1.H-2     CA.12.2.1.H-1     CO.2.1.1.H-1

CO.4.2.1.S.1     NV.1.1.2.S-1     NV.5.1.1a.S-1     NM.2.1.1.S-1

UT.1.1.2.C-2     UT.2.2.1.H-1     UT.5.1.1.S-1     WY.1.1.1.S-1

**Figure 39.** Weeping -eye and Radiant-eye designs (drawings by David Lee).

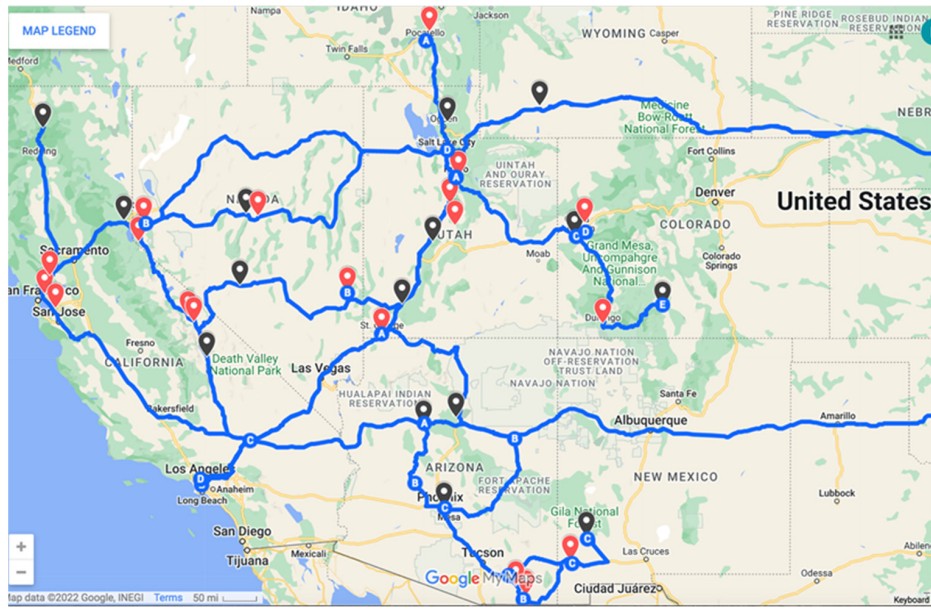

**Figure 40.** Distribution of classic weeping-eye and variant weeping-eye images is designated with black nodes.

**Funding:** This research received no external funding.

**Institutional Review Board Statement:** Not applicable.

**Informed Consent Statement:** Not applicable.

**Data Availability Statement:** Data is available from author at mleighm33@gmail.com.

**Acknowledgments:** A debt of gratitude to the many inquiring minds and co-wanderers from over the years who have accompanied this quest, beginning with Al Elsasser who opened the first door in 1983, and followed by Terry Carter, Paul Freeman, John and Mavis Greer, Judy Hilbish, Alvin McLane, Robert Messick, Andy Pate, Bill Sonin, and of course and especially, Amy Leska Marymor. A bow to the many generous individuals along the way who shared of their time, knowledge, and access to the WMP sites, among them: Bob Allen, Margaret Berrier, Jeff Bielen, Kathryn Chargin, Louise Coleville, Shawn Davies, Jeff Fentress, Robert Fisher, Jim Guymon, Jon and Sheila Harman, Charlie Harnack, Anthony Howell, Utahna and Shep Jessop, David Kaiser, Jim Keyser, Jane Kolber, David Lee, Clyde Low, Dave Manley, Tamia Marg, Bob Mark and Evelyn Billo, Al Matheson, Carol Ormsbee and Steve Schwartz, Jerry Oser, Nancy Potenza, Jerry Shaw, Leroy Unglaub and Stewart Lasseter Wyld. Appreciations, also, to my readers whose editorial comments have greatly improved this text. Any shortcomings that persist are solely my own. Thank you Kathryn Chargin, James Carroll, Susan Evans, Bill Hyder, Bill Jasper, Andy Gulliford, Dave Kaiser, Jim Keyser, Jeff LaFave, Bill Layman, Amy Leska Marymor, Robin Stratton, Curtis Whitear, and Zeese Papanikolas. I would also like to express a special debt of gratitude to David Lee for his renderings of the WMP panels. Lastly, but not least, I extend my gratitude to Sandee Pan and the editorial staff at MDPI whose expertise greatly improved the final manuscript.

**Conflicts of Interest:** The author declares no conflict of interest.

## Appendix A. Sources and Translations for Panel 2, Green River WY

**Table A1.** Source material for the transliteration of panel 2 at Green River WY.

| Graphic Image | Position [1] | Cultural Source | Reference (Citation) | Translation |
|---|---|---|---|---|
|  | 1.1 | Ojibwa | Copway (1850)<br>Mallery 10th Annual 1893<br>(Mallery 1893, p. 586/f. 853)<br>Tomkins (1926, p. 68) | Bad |
|  | 1.2 | Oglala Sioux | Mallery 10th Annual 1893<br>(Mallery 1893, p. 586/f. 853)<br>Tomkins (1926, p. 72)<br>Tehanetorens (1998, p. 34) | Winter, snow, frozen |
|  | 1.3 | Oglala Sioux | Mallery 10th Annual 1893<br>(Mallery 1893, p. 605/f. 967 and 969)<br>Tehanetorens (1998, p. 34) | Elk (hunt) |
|  | 1.4 | Ojibwa | Tehanetorens (1998, p. 14) | Deer (hunt) |
|  | 1.5 | Ojibwa | Copway (1850, p. 133)<br>Mallery 10th Annual 1893<br>(Mallery 1893, p. 586/f. 853)<br>Tomkins (1926, p. 68) | Good (the inverse of the sign for "bad") |
|  | 1.6 | Oglala Sioux | Mallery 10th Annual 1893<br>(Mallery 1893, p. 605/f. 967 and 969)<br>Tomkins (1926, p. 72)<br>Tehanetorens (1998, p. 34) | Winter, snow, frozen |

Table A1. *Cont.*

| Graphic Image | Position [1] | Cultural Source | Reference (Citation) | Translation |
|---|---|---|---|---|
| | 1.7 | Ojibwa; Sioux | Tomkins (1926, p. 73) | Running Fox (similar in form to other animal depictions, i.e., wolf, dog) |
| | 1.8 | Ojibwa Mide | Hoffman (1891, p. 260/pl. XVI.b) Mallery 10th Annual 1893 (Mallery 1893, pp. 231–54) Tomkins (1926, p. 71) | Rest; pause; stop (period; comma) |
| | 1.9 | Ojibwa Mide | Hoffman (1891, p. 291) Mallery 10th Annual 1893 (Mallery 1893, p. 240) Tomkins (1929, 3rd ed., p. 77) | Spirit surrounded by a line indicating a shore; Hidden; obscure |
| | 2.1 | Egyptian + Ojibwa; Sioux | Renouf (1875) Mallery 1st Annual 1881 (Mallery 1881, p. 361/f. 149) Mallery 10th Annual 1893 (Mallery 1893, p. 716/f. 1186) Tomkins (1926) | To go, to come, locomotion; Motion; and hunt, hunting |
| | 2.2 | Battiste Good's (Sioux) Winter Count | Mallery 4th Annual 1886 (Mallery 1886, p. 217/f. 138) Mallery 10th Annual 1893 (Mallery 1893, p. 576/f.824) Tomkins (1926, p. 71) | Sick; pain; death |
| | 2.3 | Ojibwa Mide | Hoffman (1891, p. 260/pl. XVI.b) Mallery 10th Annual 1893 (Mallery 1893, pp. 231–54) Tomkins (1926, p. 71) | Rest; pause; stop (period; comma) [2] |
| | 2.4 | Ojibwa + Red Cloud's Dakota Sioux census | Schoolcraft (1851, vol. 1/pl. 58.1) Mallery 4th Annual 1886 (Mallery 1886, pl.LXV.96) Tomkins (1926, p. 71) | Three roads |
| | 2.5 | Ojibwa; Sioux | Tomkins (1926, p. 73) | Running Fox (similar in form to other animal depictions, i.e., wolf, dog) |
| | 2.6 | Ojibwa; Sioux | Schoolcraft (1851, vol. 1/pl. 58.1) | Three |
| | 2.7 | Ojibwa; Sioux | Copway (1850, p. 132) Mallery 10th Annual 1893 (Mallery 1893, p. 660/f. 1071) Tomkins (1926, p. 70) | Life |
| | Signifier; offset bottom right | Ojibwa | Copway (1850, p. 132) Mallery 10th Annual 1893 (Mallery 1893, p. 660/f. 1071) Tomkins (1926, p. 70) | Wise Man |
| | Signifier; below | Unique | | Hand with inscribed seated horseman in the palm |

[1] Numeric designation indicates the line, followed by the images' position in the line. Lines read left to right, top to bottom. [2] These marks are lightly incised and may date to a later interaction with the WMP panel.

**Appendix B. WMP Sites and Their Historic Associations**

**Figure A1.** Four variable Venn Diagram of WMP sites' historic associations.

92% of WMP sites are co-associated with a combination of three or all four of the tested historical features dating to the period of Western Expansion. To merit the designation of "associated", the WMP site must overlook, be adjacent to, or be located proximal to the named historic feature. The Venn Diagram above illustrates that of the 15 possible combinations of four variables, the 39 WMP sites are concentrated in five combinations, only.

A = Wagon Roads and/or Railroad Rights-of-Way
B = Mines and/or Quarries
C = Town Sites
D = Mormon Towns, Routes, and/or Commerce

**Table A2.** Place specific details of WMP sites' historic associations.

| | WMP Sites Overlook, are Adjacent to, and/or are Proximal to Historic Associations | | | | |
|---|---|---|---|---|---|
| **Sites By State** | **Wagon Roads** | **Railroads** | **Mines and Quarries** | **Town Sites** | **Mormon Association** |
| AZ.1: Ash Fork | | Atlantic and Pacific RR. Santa Fe, Prescott and Phoenix RR. | Jerome Mining District. Sandstone flagstone quarries. | Ash Fork | |
| AZ.2: Bisbee | Shearer Canyon Wagon Road. | El Paso and Southeastern RR. | Bisbee Mining District. | Bisbee | Commerce: St. David lumber industry via EP and SE RR |
| AZ.3: Flagstaff | Beale Wagon Road. | Atlantic and Pacific RR. | | Flagstaff | |
| AZ.4: Tempe | Mormon Honeymoon Trail. | Maricopa and Tempe RR. Phoenix and Eastern RR. | Sandstone quarry. Wickenburg and Superstition Mtns. Mining District. | Tempe | Hayden's Butte. Honeymoon Trail. Mormon settlement. Tempe. Mesa. |
| AZ.5: Tombstone | Tombstone Stage Road. | New Mexico and Arizona Rail bed. El Paso and Southwestern RR. | Tombstone Mining District. | Tombstone | Commerce: St. David lumber industry via EP and SW RR |
| CA.1: Berkeley | Claremont Canyon/Fish Ranch Road Wagon Road. | | | Berkeley. Brooklyn Township (Oakland) | Mormon settlement. Brooklyn Township (Oakland) |
| CA.2: Berkeley | | California & Nevada RR. | | Orinda Station. Berkeley. Brooklyn Township (Oakland) | Mormon settlement. Brooklyn Township (Oakland) |
| CA.3: Berkeley | Claremont Canyon/Fish Ranch Road Wagon Road. | | | Berkeley. Brooklyn Township (Oakland) | Mormon settlement. Brooklyn Township (Oakland) |

**Table A2.** *Cont.*

| | WMP Sites Overlook, are Adjacent to, and/or are Proximal to Historic Associations | | | | |
|---|---|---|---|---|---|
| **Sites By State** | **Wagon Roads** | **Railroads** | **Mines and Quarries** | **Town Sites** | **Mormon Association** |
| CA.4: Berkeley | Claremont Canyon/Fish Ranch Road Wagon Road. | | | Berkeley. Brooklyn Township (Oakland) | Mormon settlement. Brooklyn Township (Oakland) |
| CA.5: Berkeley | Claremont Canyon/Fish Ranch Road Wagon Road. | | | Berkeley. Brooklyn Township (Oakland) | Mormon settlement. Brooklyn Township (Oakland) |
| CA.6: Birchim Canyon | Sherwin Wagon Road. Midland Trail. | | Mammoth Mining District. | Bishop | |
| CA.7: Castle Crag | Local Mule Trail. Siskiyou Trail. | Southern Pacific RR. | Dunsmuir Mining District. | Dunsmuir | |
| CA.8: Mission Pass | Mission Pass. | | | Washington Township (Fremont). | Mormon settlement. Washington Township (Fremont). |
| CA.9: Vargas Plateau | | Western Pacific RR. Transcontinental RR Terminus. | | Washington Township (Fremont). | Mormon settlement. Washington Township (Fremont). |

**Table A2.** *Cont.*

| | WMP Sites Overlook, are Adjacent to, and/or are Proximal to Historic Associations | | | | |
|---|---|---|---|---|---|
| **Sites By State** | **Wagon Roads** | **Railroads** | **Mines and Quarries** | **Town Sites** | **Mormon Association** |
| CA.10: Alabama Hills | Los Angeles Road (southern route) of the Midland Trail. | Carson & Colorado RR (aka Nevada & California RR, Southern Pacific RR). The Southern Pacific Jawbone Branch from Mojave to Owenyo passes just below the WMP site, was completed in 1910, and served the building of the L.A. Aqueduct. | Local mining adits. Keeler Mining District. | Lone Pine. Keeler. | |
| CA.11: Rockville | Old Fremont Trail. Sonoma to Benicia to Sacramento Wagon Road. | California Pacific RR. Central Pacific RR. Southern Pacific RR. | Cordelia basalt cobblestone quarry. | Rockville Corners. Cordelia. | |
| CA.12: Truckee | Emigrant Trail at Donner_Pass (Cold_Stream Canyon branch). | Union Pacific RR. | Comstock Mining District. | Truckee. | |
| CA.13: Bishop | Midland Trail. | Carson & Colorado RR (aka Nevada & California RR, Southern Pacific RR). | Local mining district (Rossi Mine, Bishop mine, etc.) | Bishop. | |
| CO.1: Cameo | Local horse trail. | | Palisades Coal Mining District. | Cameo. Palisades. | |
| CO.2: Del Norte | Summitville Mine wagon road. | Denver and Rio Grande RR. | Summitville Mine Road. | Del Norte | Commerce: Local Mormon population at Almarosa supplied labor to Summitville mine and the D&RG RR. |

**Table A2.** *Cont.*

| | WMP Sites Overlook, are Adjacent to, and/or are Proximal to Historic Associations | | | | |
|---|---|---|---|---|---|
| **Sites By State** | **Wagon Roads** | **Railroads** | **Mines and Quarries** | **Town Sites** | **Mormon Association** |
| CO.3: Durango | Animas Canyon Toll Road. | Silverton & Durango RR (D&RG RR) | Silverton Mining District rail access. | Durango | |
| CO.4: Grand Junction | Monument Valley Wagon Road. | Denver and Rio Grande RR (Denver and Rio Grande Western RR). | Portneuf Valley Gold Mining District. | Grand Junction | |
| ID.1: Pocatello | Oregon Trail. | Utah & Northern RR (aka Oregon Short Line; later Union Pacific RR). | Reese River Mining District. | Pocatello | Bannock Valley Mormon Settlement. Utah Northern RR funded by John Young, Brigham Young's son (later Utah & Northern Railway). |
| NV.1: Austin | Overland Trail. | Nevada Central Railway. Austin City Railway. | | Austin. | |
| NV.2: Genoa | Kingsbury Grade (aka Georgetown Trail, Dagget Pass Trail and Pass. Overland Emigrant Trail. | | Supply depot to Comstock Lode, Bodie, Tonopah, Goldfield Mining Districts. | Genoa. | Established as a Mormon Settlement by Brigham Young. |
| NV.3: Hickison Summit | Overland Trail | | | | |
| NV.4: Pioche | Concorde coach roads to Palisade via Hamilton in White County, and to Salt Lake. | Three railroads were organized to build lines into Pioche—including the Salt Lake, Sevier Valley & Pioche Railroad (which was a Mormon line) and the Palisade, Eureka & Pioche. | Pioche Mining District. | Pioche. | William Hamblin, a Latter Day Saint missionary, was led to silver deposits in the vicinity of Pioche by a Native American Paiute. |
| NV.5: Tonopah | Midland Trail. | Tonopah RR (later Tonopah and Goldfield RR). | Tonopah Mining District. | Tonopah | |

**Table A2.** *Cont.*

| | WMP Sites Overlook, are Adjacent to, and/or are Proximal to Historic Associations | | | | |
|---|---|---|---|---|---|
| **Sites By State** | **Wagon Roads** | **Railroads** | **Mines and Quarries** | **Town Sites** | **Mormon Association** |
| NV.6: Virginia City | Geiger Toll Road. | Virginia and Truckee RR. Carson and Colorado RR (later Nevada & California RR, aka Southern Pacific). | Comstock Lode Mining District. | Virginia City | |
| NM.1: Lordsburg | Butterfield Trail. | Southern Pacific RR. | Shakespeare Mining District. | Lordsburg | |
| NM.2: Silver City | Chloride Flat Wagon Road. | Silver City, Deming and Pacific RR (later Santa Fe RR). | Red Hills Mine | Silver City | |
| UT.1: Cedar City | California Mormon Trail. | | Iron mining. | Cedar City | Mormon settlement. California Mormon Trail. |
| UT.2: Fillmore | Chalk Creek Canyon (sandstone quarry) Road. California Mormon Trail. | | Sandstone quarry. | Fillmore | Mormon settlement. California Mormon Trail. |
| UT.3: Manti | Pioneer trail_ from Nephi to Manti (along highway 128 corridor). | Sanpete Valley Railway. Denver and Rio Grande Western. | Oolite limestone quarry used in construction of Manti Temple. | Manti | Manti Mormon Temple site. Mormon Pioneer Trail. |
| UT.4: Nephi | California Mormon Trail. | Utah and Southern RR. Sanpete Valley RR. | Supply hub (coal, milling, agriculture) to Wales Mining District. | Nephi | Mormon settlement. California Mormon Trail. |
| UT.5: Ogden | California Mormon Trail. | Utah and Northern RR. | Supply hub (coal) to Transcontinental RR. | Ogden | Mormon settlement. California Mormon Trail. |

**Table A2.** *Cont.*

| | WMP Sites Overlook, are Adjacent to, and/or are Proximal to Historic Associations | | | | |
|---|---|---|---|---|---|
| **Sites By State** | **Wagon Roads** | **Railroads** | **Mines and Quarries** | **Town Sites** | **Mormon Association** |
| UT.6: Provo | California Mormon Trail. | Utah and Northern RR. | | Provo | Mormon settlement. California Mormon Trail. |
| UT.7: St George | California Mormon Trail. | | Mormon Temple sandstone quarry. Mormon Temple basalt quarry. | St George | Mormon settlement. California Mormon Trail. Mormon Temple site. |
| WY.1: Green River | Mormon Trail. Emigrant Trail. | Union Pacific RR. | | Green River. | Mormon Trail. |

## Notes

1    (Marymor 2020, 2021a; Marymor and Marymor 2015, 2017; and Marymor and Lanman 2021) are all prior publications by the author of this paper and are directly relevant for further background on the WMP topic.

2    Translations of WMP panels are derived from the method discussed in Section 6 of this paper.

3    Including California, Arizona, Nevada, Utah, New Mexico, Colorado, Wyoming, and Idaho (Figure 2).

4    For grammatical convenience, this study uses the terms "author" and "he" to refer to the creator of the WMP engravings. Authorship of the WMPs may be the responsibility of one or more persons, and the gender of the author, or authors, is also unknown.

5    Originally named "Dixie".

6    Including, Ash Fork and Bisbee, Arizona, Pocatello, Idaho, and Birchim Creek, Truckee, and Dunsmuir, California.

7    *Reports of explorations and surveys, to ascertain the most practicable and economical route for a railroad from the Mississippi River to the Pacific Ocean:* United States. War Dept., Henry, Joseph, 1797–1878., Baird, Spencer Fullerton, 1823–1887., United States. Army. Washington: A.O.P. Nicholson, printer {etc.], 1855–1860.

8    Ibid. See also W.J. Hoffman (1891). 7th Annual Report to the Bureau of Ethnology, Smithsonian Institution, pp. 186–87; referenced in Mallery (1893, pp. 231–54).

9    Garrick Mallery (1877). A Calendar of the Dakota Indians. 1(3):1–26.

10    See Judy Hilbish (2017) for descriptions of her early searches of ethnographic literature as source material for Western Message Petroglyh imagery. Although not in agreement that Mallery's work is central to the WMP project, Hilbish was the first to turn to ethnographic literature in an attempt to unravel the mystery of WMPs.

11    This image is reproduced in Mallery (1893) in larger format as a stand-alone image with generally the same discussion as appeared in Mallery's (1886) 4th annual report.

12    William Tomkins (1926). *Universal Indian Sign Language*. Preface.

13    Referred to as "Southern Cult" in Rands (1957).

14    The Ojibwa Medēwiwin image is completely different in form, but shares the convention of lines descending from the eyes (Schoolcraft 1851, v. I, pl. 54, Figure 27).

15    Here, Mallery is referring to lines emanating downward from the eye. Mallery repeats this discussion in the 10th Annual Report (Mallery 1893, p. 642).

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
