# Peer review of "Western Message Petroglyphs: A Faux Indian Picture-Writing Project in the American West"

_arts, 2022_

Round 1

Reviewer 1 Report

Some of the figures are not intelligible at the scale presented. They may need to be rotated as full page images, and/or reformatted.

I must admit that I cannot comprehend the motives of the WMP author(s) nor the meanings of the 'translations'. Was it intended as a puzzle?

Reference to "Stone Diary: Solving the Mystery of the Western Message Petroglyphs" needs to be included.

Author Response

Dear Peer-Reviewer #1:

Thank you for your comments.

The layout, including size, orientation, and placement, of figures, maps and tables will be addressed in cooperation with the Editor prior to publication.

Regarding the translations:  Only the transliterated definitions of the picture symbols can be ascertained from the historic record, and I note that not every symbol has thus far been located in the historic record. The translations, as described in the text, are a poetic leap that is grounded in the transliterations that were  sourced from the historic record, to the extent we have it. To date, there is no verifiable means to ascertain that the translations offered are accurate and reflective of the author's original intent. That said, it would appear that the author's selection of semi-remote locations and esoteric picture-writing symbols reflect his intent to obscure and restrict access to his meanings. His audience may have been intended for a small group in-the-know, or perhaps were private meditations. Restricted access to knowledge fits comfortably with the proposal that this author had roots in Mormonism, fraternal organizations, or both - both institutions sharing the value of restricting access to esoteric knowledge. Rather than espousing Mormon ideology, the themes and tenor (philosophic, ironic, and numinous) suggest a counter-intellect, one with an over-all sympathy for human nature and the human condition. I have expressed these thoughts in my narrative, and ask if you would like to suggest that more should be said on this topic?

Thank you for pointing out the unintended oversight in regard to Judy Hilbish's publication. It will be included in the final draft.

Regards,

Reviewer 2 Report

I think this is a great paper! The hypothesis to the conclusion is very well threaded with comprehensive narration. 

Author Response

Dear Peer-Reviewer #2,

Thank you for your comments. Would you be able to be more specific for the following three areas that you indicated could be improved?

Are the research design, questions, hypotheses and methods clearly stated? ( ) (x) ( ) ( )
Are the arguments and discussion of findings coherent, balanced and compelling? ( ) (x) ( ) ( )

Thanks in advance,